# Chromosomal genome assembly resolves drug resistance loci in the parasitic nematode *Teladorsagia circumcincta*

**Jennifer McIntyre**[1]*, **Alison Morrison**[2], **Kirsty Maitland**[1], **Duncan Berger**[3], **Daniel R. G. Price**[2], **Sam Dougan**[3], **Dionysis Grigoriadis**[4], **Alan Tracey**[3], **Nancy Holroyd**[3], **Katie Bull**[5], **Hannah Rose Vineer**[5,6], **Mike J. Glover**[7], **Eric R. Morgan**[5,8], **Alasdair J. Nisbet**[2], **Tom N. McNeilly**[2], **Yvonne Bartley**[2], **Neil Sargison**[9], **Dave Bartley**[2], **Matt Berriman**[3¤a], **James A. Cotton**[3¤b], **Eileen Devaney**[1], **Roz Laing**[1], **Stephen R. Doyle**[3]*

**1** School of Biodiversity, One Health and Veterinary Medicine, College of Medical, Veterinary and Life Sciences, University of Glasgow, Garscube Campus, Glasgow, United Kingdom, **2** Moredun Research Institute, Pentlands Science Park, Bush Loan, Penicuik, United Kingdom, **3** Wellcome Sanger Institute, Hinxton, Cambridgeshire, United Kingdom, **4** European Molecular Biology Laboratory, European Bioinformatics Institute, Hinxton, Cambridgeshire, United Kingdom, **5** Veterinary Parasitology and Ecology Group, University of Bristol, Bristol, United Kingdom, **6** University of Liverpool, Institute of Infection, Veterinary and Ecological Sciences, Leahurst Campus, Neston, Cheshire, United Kingdom, **7** Torch Farm & Equine Ltd., Veterinary Surgeons, South Molton, Devon, United Kingdom, **8** Queen's University Belfast, School of Biological Sciences, Belfast, United Kingdom, **9** Royal (Dick) School of Veterinary Studies, University of Edinburgh, Edinburgh, United Kingdom

¤a Current address: School of Infection and Immunity, College of Medical, Veterinary and Life Sciences, University of Glasgow, Glasgow, United Kingdom
¤b Current address: School of Biodiversity, One Health and Veterinary Medicine, College of Medical, Veterinary and Life Sciences, University of Glasgow, Glasgow, United Kingdom
* Jennifer.McIntyre@glasgow.ac.uk (JM); stephen.doyle@sanger.ac.uk (SRD)

**Data Availability Statement:** Sequencing data generated for this study are available from the

## Abstract

The parasitic nematode *Teladorsagia circumcincta* is one of the most important pathogens of sheep and goats in temperate climates worldwide and can rapidly evolve resistance to drugs used to control it. To understand the genetics of drug resistance, we have generated a highly contiguous genome assembly for the UK *T. circumcincta* isolate, MTci2. Assembly using PacBio long-reads and Hi-C long-molecule scaffolding together with manual curation resulted in a 573 Mb assembly (N50 = 84 Mb, total scaffolds = 1,286) with five autosomal and one sex-linked chromosomal-scale scaffolds consistent with its karyotype. The genome resource was further improved via annotation of 22,948 genes, with manual curation of over 3,200 of these, resulting in a robust and near complete resource (96.3% complete protein BUSCOs) to support basic and applied research on this important veterinary pathogen. Genome-wide analyses of drug resistance, combining evidence from three distinct experiments, identified selection around known candidate genes for benzimidazole, levamisole and ivermectin resistance, as well as novel regions associated with ivermectin and moxidectin resistance. These insights into contemporary and historic genetic selection further emphasise the importance of contiguous genome assemblies in interpreting genome-wide genetic variation associated with drug resistance and identifying key loci to prioritise in developing diagnostic markers of anthelmintic resistance to support parasite control.

European Nucleotide Archive (ENA); individual sample information, including sample and run accession numbers, are described in S2 Table. The genome is available under the ENA Project accession PRJEB80725. https://www.ebi.ac.uk/ena/browser/view/PRJEB80725 The code used to analyse raw data and generate figures is available from GitHub (https://github.com/stephenrdoyle/tcircumcincta_genome).

**Funding:** This work was supported by the Biotechnology and Biological Sciences Research Council (BBSRC) [BBN50385X/1 to ED and BB/M003949 to ED], AHDB Beef and Lamb and the KTN (6130010011 to JM and ED), the Scottish Government's Rural and Environment Science and Analytical Services (RESAS) division through Underpinning National Capacity and award MRI-A2-6 (to DB), Meat and Livestock Australia (MLA) project P.PSH.1445 (to AN), the Wellcome Trust (208390/Z/17/Z (ED), 206194 (MB), 220540/Z/20/A (SRD), and 216614/Z/19/Z to RL), and UKRI (MR/T020733/1 to SRD). The funders had no role in study design, data collection and analysis, decision to publish, or preparation of the manuscript.

**Competing interests:** The authors have declared that no competing interests exist.

## Author summary

Understanding the genetics of anthelmintic resistance is a critical part of the sustainable control of parasitic worms. Here, we have generated a chromosome-scale genome assembly for the gastrointestinal nematode *Teladorsagia circumcincta*, one of the most important pathogens of sheep and goats in temperate climates worldwide. This genome and its annotation offer a substantial improvement over existing genetic resources and will enable new insight into the biology of this parasite and new opportunities to characterise therapeutic targets such as drug and vaccine candidates. We used this resource to map genetic variation associated with resistance to multiple anthelmintic drug classes used as the primary means of parasite control, confirming known candidate genes and variants (i.e., *beta-tubulin isotypes 1* and *2* associated with benzimidazole resistance and *acr-8* associated with levamisole resistance, and *pgp-9* copy number variation, of which overexpression is associated broadly with anthelmintic resistance) and revealing new regions of the genome associated with drug treatment responses (i.e. 34–38 Mb on chromosome 5 associated with ivermectin resistance). Our results highlight the importance of a contiguous genome assembly and the use of complementary experimental approaches to reveal the impact of drug-mediated selection, both historically and directly in response to treatment, on genome-wide genetic variation.

## Introduction

Parasitic worms can rapidly evolve to resist the anthelmintic drugs used to control them. Anthelmintic resistance is a global problem and places a significant economic and animal health burden on livestock industries and the care of domestic animals [1,2]. Furthermore, resistance threatens the large-scale control programmes that almost exclusively rely on anthelmintic drugs to manage human-infective helminths as a public health problem [3,4]. Understanding how resistance emerges is critical to the sustainable control of helminth infections. As such, significant efforts have been made to define the genetic drivers of anthelmintic resistance. Despite much progress, critical information gaps prevent the use of genetic-based diagnostic tests to inform and support control efforts.

Recent genomic approaches have begun to shed light on the challenges of defining causal variants of anthelmintic resistance. For example, genomic studies have revealed extensive genetic diversity within and differentiation between helminth populations [5–9], highlighting that the vast majority of genetic variants between susceptible and resistant isolates have little to do with resistance, but are due to distinct evolutionary histories of parasite populations that are not freely mating [10]. These genomic analyses have also begun to differentiate between resistance mechanisms consistently driven by the same single or multiple genes and those arising from either shared or distinct genetic backgrounds to produce the same apparent drug resistance phenotype [11,12]. Moreover, studies using the model nematode *Caenorhabditis elegans* have demonstrated clear examples where natural background genetic variation within resistance-defined genes led to variation in resistance phenotypes [13,14]; such background variation in the context of resistance is yet to be widely defined in parasitic nematodes but is almost certain to exist. These genome-wide perspectives can, at least in part, explain why genetic comparisons of candidate genes between susceptible and resistant isolates can be misleading. High-quality genetic resources are vital for interpreting genome-wide analyses but currently cover only a select number of helminth species [15]. Chromosome-scale genome assemblies have been key to understanding the impact of the selection of genetic variation

throughout the genome, supporting previous candidate genes and revealing new genes associated with resistance. This is perhaps best illustrated by studies investigating anthelmintic resistance in the gastrointestinal nematode *Haemonchus contortus* (resistance to ivermectin, levamisole, benzimidazole, or monepantel)[16–20]. High-quality, contiguous genome assemblies have also been instrumental in providing new insights into resistance variants and mechanisms in *Fasciola hepatica* (triclabendazole resistance)[21] and *Dirofilaria immitis* (ivermectin resistance) [22], as well as the human-infective *Schistosoma mansoni* (praziquantel resistance) [23] and *Onchocerca volvulus* (ivermectin resistance) [24].

The brown-stomach worm *Teladorsagia circumcincta* (previously known as *Ostertagia circumcincta*, with morphotypes *T. davtiani* and *T. trifurcata* [25]) is an obligate gastrointestinal parasite of small ruminants and is one of the most important pathogens of sheep and goats in temperate climates around the world. Infections are a significant economic and animal health concern [26]; the resultant abomasal inflammation leads to clinical signs, including a reduction in appetite, reduced weight gain or weight loss, and diarrhoea, and can lead to death in heavily infected animals. Anthelmintic resistance to all major broad-spectrum classes of drugs is widespread. To understand the genetics of resistance, a draft genome assembly (WASHU assembly) of a semi-inbred isolate from New Zealand was sequenced and used to assess genetic variation from a backcross between a susceptible isolate ($S_{inbred}$) and a triple-resistant field isolate ($RS^3$; ivermectin, levamisole, and benzimidazole resistant)[27]. The authors identified several candidate genes associated with resistance to each anthelmintic class and, despite the highly fragmented genome, concluded that multi-drug resistance was polygenic. The draft genome was further scaffolded using additional data derived from an Australian isolate, resulting in significantly greater contiguity than the original assembly (DNAZOO assembly); however, some conserved genes were lost, and only approximately half of the genome was assembled into scaffolds that, although large, were smaller than the expected chromosome sizes [28].

Here, we have generated a chromosome-scale genome assembly for the anthelmintic susceptible UK MTci2 isolate of *T. circumcincta* and used it to identify loci associated with resistance to three distinct classes of globally used anthelmintic drugs. Our assembly of a single isolate, generated using PacBio long-read sequencing and Hi-C scaffolding, together with extensive manual curation of the genome and annotation, offers a significant improvement in contiguity and completeness relative to existing assemblies and represents a robust resource for this important global pathogen. Comparative population genetic analyses of new and existing datasets provide a genome-wide perspective of the impact of contemporary and historic drug selection, further refining our understanding of the genetics of drug resistance and prioritising key loci for developing diagnostic markers to support parasite control.

## Results

### The chromosomal-scale genome assembly of *Teladorsagia circumcincta* MTci2

We used PacBio long-read sequencing with Hi-C scaffolding to generate a highly contiguous genome assembly for the UK-derived MTci2 isolate of *T. circumcincta* (tci2_wsi3.0; ENA accession: ERZ24880405; **Table 1**). Spanning 573 Mb, 86.6% of the assembly is scaffolded into five autosomes, a sex-linked X chromosome, and a mitochondrial genome (**Fig 1A**). Chromosomal identities have been assigned based on comparison to the chromosomal assembly of the closely related clade V gastrointestinal helminth *H. contortus* (**Fig 1B**). The remaining 13.6% of the assembly has been conservatively retained; these sequences have either been assigned to (i) chromosomal-linkage groups, which could not be confidently placed within the

**Table 1. Genome and annotation statistics of *Teladorsagia circumcincta* assemblies and close relatives.**

| Genome assembly | Assembly size (Mb) | Scaffolds (n) | N50 (Mb) | N90 (Mb) | Largest (Mb) | Gaps (n) | Bases Missing, Ns (Mb) [a] | Genes (n) | mRNAs (n) | Complete BUSCOs (%) Genome [b] | Complete BUSCOs (%) Protein [c] |
|---|---|---|---|---|---|---|---|---|---|---|---|
| *T. circumcincta* tci2_wsi3.0 | 573.0 | 1,286 | 84.0 | 2.4 | 94.8 | 10,977 | 3.6 | 22,948 | 36,039 | 85.2 | 96.3 |
| *T. circumcincta* WASHU | 700.6 | 81,734 | 0.047 | 0.002 | 1.5 | 131,657 | 124.5 | 25,572 | 25,570 | 67.4 | 40.2 |
| *T. circumcincta* DNAZOO [c] | 593.2 | 52,507 | 57.1 | 0.004 | 66.6 | 120,667 | 111.8 | 28,082 | 30,055 | 65.8 | na |
| *H. contortus* ISE V4 | 283.4 | 7 | 47.4 | 43.6 | 51.8 | 185 | 5.7 | 19,778 | 21,320 | 83.5 | 96.2 |
| *C. elegans* [d] | 100.3 | 7 | 17.5 | 13.8 | 20.9 | 0 | 0 | 18,178 | 32,332 | 98.8 | 100.0 |

a. Bases missing, Ns refers to bases introduced into the assembly by scaffolding two contigs; the number of Ns introduced depends on the scaffolding approach that differs between assemblies. These Ns form a fraction of the total assembly size (column 1) and, therefore, provide an estimate of the proportion of the genome that is missing.

b. Genome completeness was assessed using BUSCO (version 5.6.1) with the Nematoda lineage dataset.

c. The number of genes and transcripts for the DNAZOO assembly were obtained from the original publication as reported by the authors, as no annotation was publicly available. Because of this, protein BUSCO scores for the protein sequences were not determined.

d. *Caenorhabditis elegans* gene count is based on "protein_coding_primary_transcript" from its GFF annotation from WormBase ParaSite.

chromosomal scaffolds, (ii) unique sequences that could not be placed, or (iii) sequences containing partially duplicated and partially unique sequences that likely represent misassemblies; however, due to the presence of unique sequence, they have been kept in the assembly.

Two existing genome assemblies for *T. circumcincta* are publicly available: WASHU [27] and DNAZOO [28]. Comparatively, our assembly represents a significant improvement over both assemblies in contiguity (**Fig 1C**) and gene content (**Tables 1 and S4**), as well as a more comprehensive representation of the repetitive content (**S5 Table**). The latter is, in part, reflected in the larger chromosome-scale scaffolds in our MTci2 assembly (longest: 94.8 Mb) relative to the DNAZOO assembly, of which only ~59% of the total assembly was assigned to chromosomal-scale scaffolds (longest: 66.6 Mb). Nonetheless, our assembly and the DNAZOO assembly showed considerable consistency in chromosome structure despite being derived from different parasite isolates and independently assembled (**Fig A in S1 Appendix**).

Annotation and manual curation of the tci2_wsi3.0 assembly identified 22,948 genes and 36,039 transcripts, representing a nearly complete (96.3%) complement of protein-coding genes, comparable with other highly curated nematode genome assemblies (**Tables 1 and S4**). The annotation contains high levels of duplicated transcripts (BUSCO: 45.5%), primarily due to the large number of annotations with multiple predicted mRNA transcripts or isoforms per gene. However, even after accounting for multiple isoforms, BUSCO duplication rates remain higher than expected (13.6% duplications after selecting the longest isoform per gene). Although some duplicated BUSCOs could be true duplications within the *T. circumcincta* genome, the higher-than-expected duplication is almost certainly due to the retention of partially duplicated sequences in the assembly, as described above. Nonetheless, we predict there should be approximately 20,000 genes, a count significantly lower than previous estimates from the draft *T. circumcincta* genomes but consistent with the curated *C. elegans* and *H. contortus* genomes. Finally, a comparison of shared one-to-one orthologs revealed 1,762 more orthologs between tci2_wsi3.0 and *C. elegans* (n = 6,137) relative to WASHU and *C. elegans* (n = 4,375); interestingly, the largest number of 1-to-1 orthologs (n = 8,461) were shared between tci2_wsi3.0 and *H. contortus* (2,273 more than between the two *T. circumcincta*

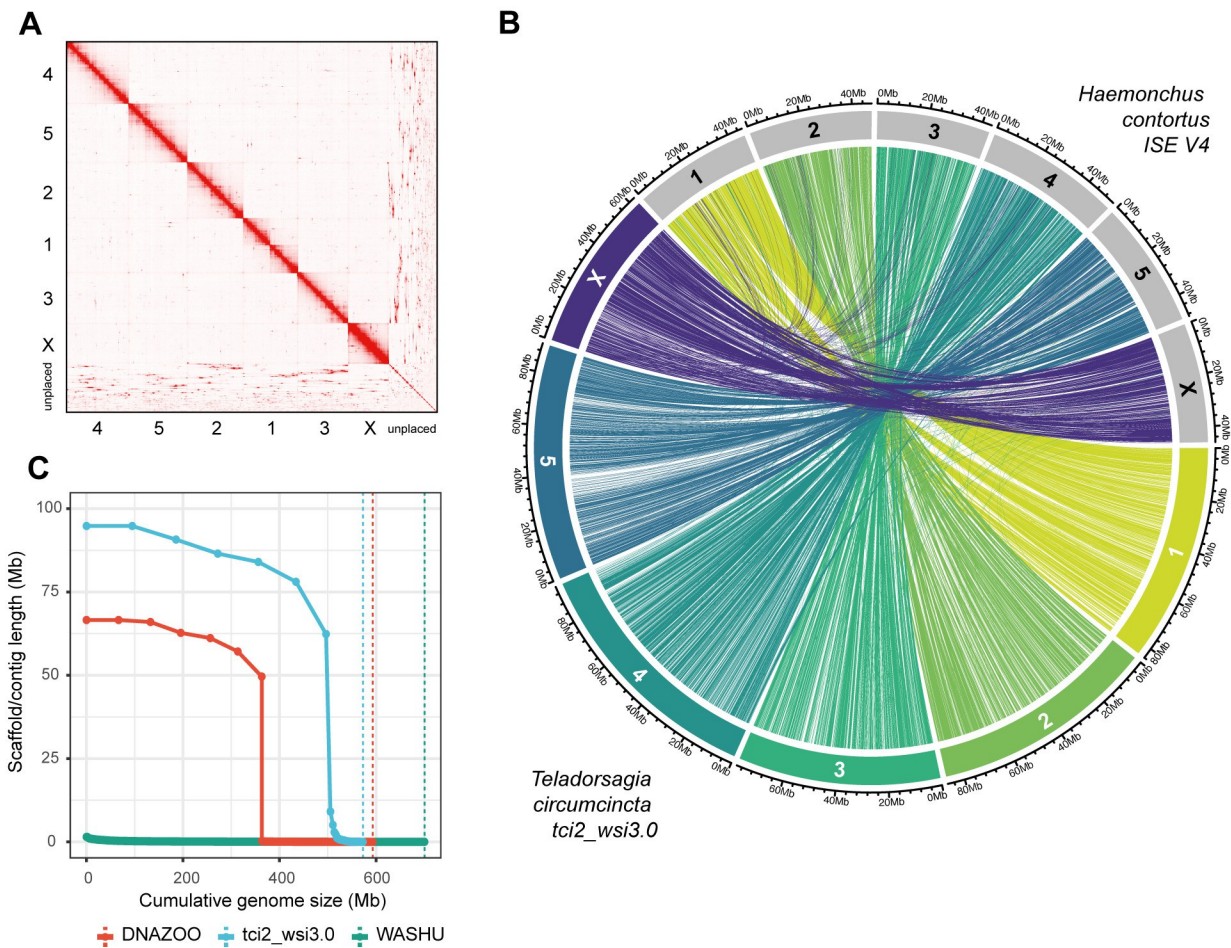

**Fig 1. Chromosome-scale assembly of the *Teladorsagia circumcincta* MTci2 isolate and comparison to related assemblies. A.** Hi-C contact map of the tci2_wsi3.0 assembly reveals the six chromosome-scale scaffolds (autosomes 1–5, X sex chromosome, ordered from largest to smallest), which have been named based on synteny with *Haemonchus contortus* chromosomes. The unplaced sequences are also shown and provide hints as to where these sequences may be located with respect to the chromosomes. **B.** Conservation of chromosome structure between tci2_wsi3.0 and *H. contortus* assemblies. The outer circle is divided into chromosomal segments, proportionally sized and coloured by tci2_wsi3.0 chromosome ID, whereas *H. contortus* segments are grey. The line segment shows positional information of 1-to-1 orthologs shared between the two genomes. **C.** A comparison of scaffold size relative to assembly length of the *T. circumcincta* genome assemblies WASHU (green), DNAZOO (red), and our tci2_wsi3.0 (blue) assembly. Each point represents an individual scaffold, and the vertical dashed line represents the total genome size for each assembly.

genomes), illustrating that the chromosomal assemblies are more biologically representative than the draft genome (**S6 Table**).

## Genome-wide mapping of drug-resistance-associated variation

The availability of a new, largely complete genome allowed us to characterise the distribution of genetic variation associated with resistance throughout the genome. We have analysed whole-genome sequencing data from three distinct experimental approaches: (i) comparison of pooled larvae sampled before and after ivermectin treatment from the field in the UK (Farm 1 and Farm 2), (ii) comparison of pooled adult susceptible ($S_{inbred}$) and backcrossed multi-drug resistant ($RS^3$; ivermectin, levamisole, and benzimidazole resistant) New Zealand isolates from Choi and colleagues [27], and (iii) comparison of groups of pools of worms from UK isolates with different, previously defined resistance phenotypes (MTci1 [susceptible, but

suspected low-level benzimidazole resistance], MTci5 [ivermectin, levamisole, and benzimid-azole resistant], and MTci7 [ivermectin, levamisole, benzimidazole, and moxidectin resistant]) to the drug-susceptible isolate MTci2. Comparison of genetic diversity between isolates revealed distinct differences between experimental groups, with the least variation between the farm isolates, but the greatest variation both between the Choi isolates and between the Choi and all other isolates (**Fig B in S1 Appendix**).

Genetic change in larvae sampled before and after ivermectin treatment of sheep on two UK farms reflects an immediate response in the adult population directly exposed to drug selection. Comparison of genetic differentiation between pre- and post-treatment on Farm 1 (**Fig 2A;** top panel) and Farm 2 (**Fig 2A;** middle panel) both revealed a single, overlapping peak of differentiation on chromosome 5 (peak ranges: 32.5–40 Mb and 35–40 Mb on Farm 1 and Farm 2, respectively), suggesting a common genetic response to ivermectin treatment. Both farms had diagnosed resistance but had different ivermectin efficacies by FECRT (**Fig C in S1 Appendix and S1 Table;** Farm 1 $FECR_{ivermectin(Tc)}$ = 81.9%, Farm 2 $FECR_{ivermectin(Tc)}$ = 22.1%); however, these differences were not reflected by the peak height in the genetic data. Comparison of the post-treatment populations from Farm 1 and Farm 2 revealed a more complex history of genetic change and likely drug-mediated selection (**Fig 2A;** bottom panel); while the peak of differentiation on chromosome 5 was apparent (peak range: 37.5–42.5 Mb), additional peaks were observed, one on chromosome 1 (peak range: 60–65 Mb), two on chromosome 4 (peak ranges: 65–72 Mb and 85–90 Mb), a second peak on chromosome 5 (peak range: 72.5–76 Mb) and a peak on chromosome X (peak range: 20–22.5 Mb). These additional peaks likely reflect previous selection by drug treatment or other pressures that acted independently on the two farm populations, considering they were also present in a comparison of pre-treatment populations between farms (**Fig D in S1 Appendix**). For example, the peak on chromosome 1 contains the *beta-tubulin isotype 1* gene that will have undergone strong selection in the past to benzimidazole treatment, to which resistance is present on both farms (**S1 Table;** Farm 1 $FECR_{benzimidazole}$ = 87.7%, Farm 2 $FECR_{benzimidazole}$ = 4.5%). Hence, these data reveal evidence of selection acting across different time scales and in response to different pressures.

Reanalysis of a genetic comparison between a susceptible isolate and a backcrossed, multi-drug resistant isolate provides a short-term, multi-generational view of intense co-selection throughout the genome (**Fig 2B**). Choi and colleagues originally argued that resistance is polygenic, with many signals of genetic differentiation observed in the context of the draft, fragmented genome assembly [27]. Here, using a highly contiguous assembly, the fragmented signals of genetic differentiation are more defined and contiguous; however, they are still challenging to interpret entirely in isolation. For example, a discrete peak associated with the *beta-tubulin isotype 1* gene is evident (peak range: 60–67.5 Mb; gene position: 62.287–62.292 Mb) when placed in the context of the UK farm data (**Fig 2A;** bottom panel); however, the overall region of differentiation in the Choi data is much broader spanning almost half of the chromosome (peak range: 30–80 Mb). Similarly, the discrete peak on chromosome 5 identified in the farm data is likely found in the Choi data (peak range: 30–42.5 Mb); however, there is also broad-scale genetic differentiation throughout the chromosome. A peak on chromosome 4 (peak range: 72–90 Mb), overlaps with the 85–90 Mb peak in the between-farm comparison data, and may be divided into two peaks of 72–85 Mb and 85–90 Mb; however, new peaks were also identified, including one on chromosome 2 (peak range: ~60–67.5 Mb, which contains *beta-tubulin isotype 2* (TCIR_1005973; gene position: 59.147–59.153 Mb), but with broader differentiation from ~25–67.5 Mb, and which corresponds with a small, suggestive peak in both the Farm 2 Pre:Post and between-farm comparisons), one on the end of chromosome 5 (peak range: 87.5–91 Mb) and one on the X chromosome (peak range: 1.5–5 Mb). There is also differentiation at the beginning of chromosome 5, which is not easily defined.

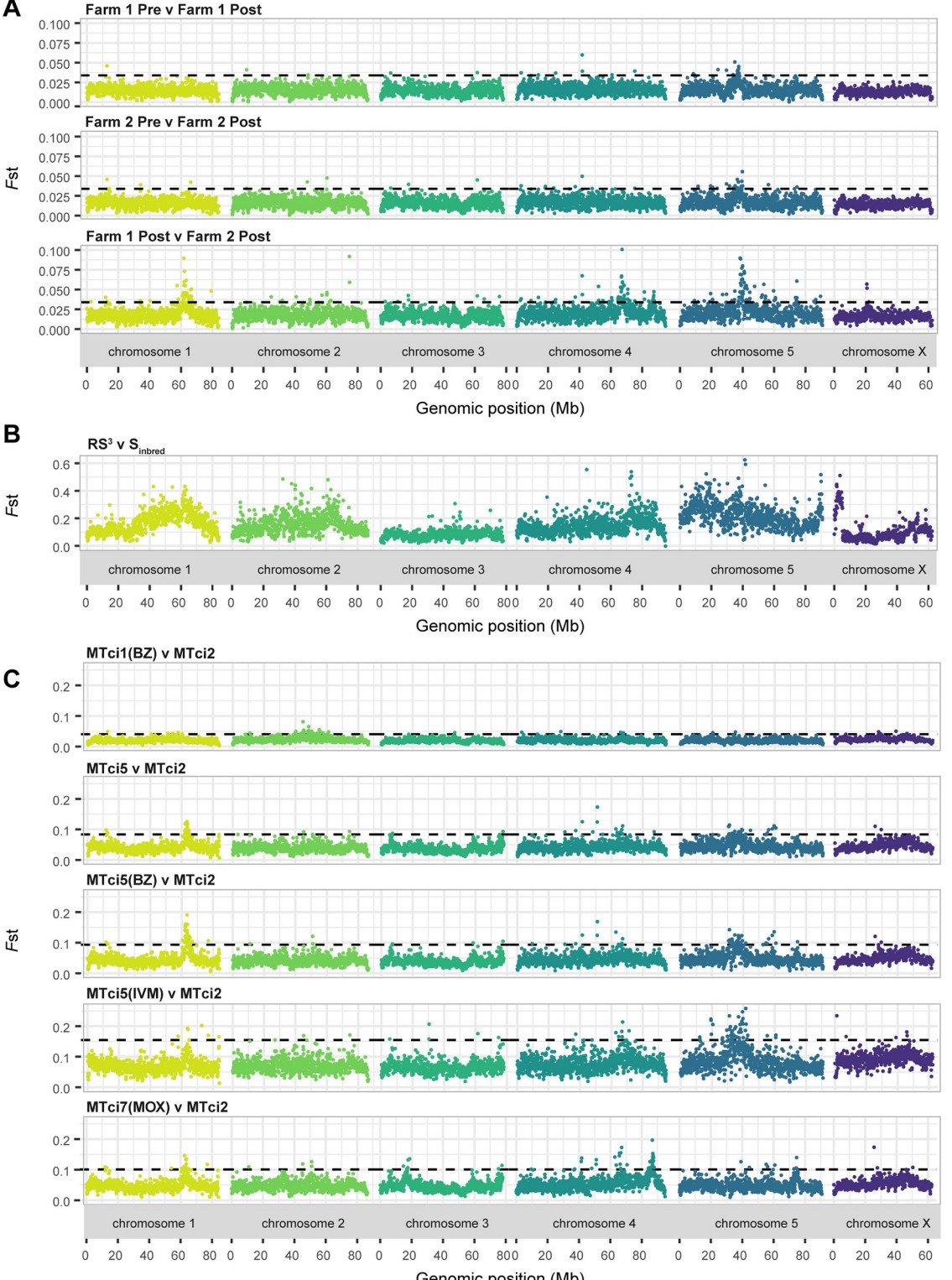

**Fig 2. Genome-wide variation associated with drug resistance.** In all plots, each point represents a measure of genetic differentiation (*F*st) calculated between pairs of samples in 100 kb windows along the genome. **A.** Comparison of genetic differentiation among pools of L3 sampled before and after ivermectin treatment on two UK farms. Top panel: Pre-vs-post ivermectin treatment on Farm 1; Middle panel: Pre-vs-post ivermectin treatment on Farm 2; Bottom panel: Farm 1-post vs Farm 2-post treatment. The dashed line represents a genome-wide level of significance, defined as the mean + 5 standard deviations of the

genome-wide *F*st calculated from technical replicates of the Farm 2 pre-treatment group (**Fig E in S1 Appendix**). **B**. Comparison of genetic differentiation between pools of adult males from an inbred susceptible population ($S_{inbred}$) and a backcrossed, multidrug-resistant population ($RS^3$) originally described by Choi and colleagues [27]. **C**. Comparison of genetic differentiation between different isolates of *T. circumcincta* with known drug resistance profiles. Top panel: MTci1 vs MTci2. Second panel: MTci5 vs MTci2. Third panel: MTci5(post-benzimidazole) vs MTci2. Fourth panel: MTci5(post-ivermectin) vs MTci2. Fifth panel: MTci7 vs MTci2. Sample details, including life stage composition, are described in **S2 Table**. The dashed line represents a genome-wide level of significance, defined as the mean + 3 standard deviations of the genome-wide *F*st per comparison.

Finally, a comparison of distinct UK isolates phenotyped for resistance provides a view of selection likely occurring independently over many parasite generations (**Fig 2C**). Comparison between MTci1 and MTci2 revealed only slight variation throughout the whole genome, even at the suspected *beta-tubulin isotype 1* locus on chromosome 1 (**Fig 2C,** top panel); both isolates are suspected of having low-level benzimidazole resistance but are largely phenotypically susceptible [29]. Three comparisons were made between MTci2 (susceptible) and different MTci5 (triple resistant; ivermectin, levamisole and benzimidazole) pools: (i) untreated (**Fig 2C,** 2nd panel), (ii) post-benzimidazole treatment (**Fig 2C,** middle panel), and (iii) post-ivermectin treatment (**Fig 2C,** 4th panel). Comparison across all three panels reveals common peaks of differentiation, including chromosome 1 (single peak), chromosome 3 (three or four suggestive peaks, most obvious in the non-ivermectin selected comparisons), and three on chromosome 5, with a possible peak on chromosome 4. The latter varies considerably (~60–70 Mb in MTci2:MTci5 and MTci2:MTci5-BZ, but broader in the MTci2:MTci5-IVM comparison (~60–80 Mb, as part of a generally noisier background), and may be shared by the between-farm comparison (65–72 Mb), and potentially the Choi comparison (70–90 Mb, greatest differentiation ~73 Mb). The chromosome 1 peak (containing the *beta-tubulin isotype-1* gene) was most prominent in the post-benzimidazole-treated comparison, whereas the main chromosome 5 peak (shared by the Farm and Choi analyses) and suspected chromosome 4 peak were most prominent in the post-ivermectin-treated comparison. Finally, a comparison of MTci7 and MTci2 revealed the same peaks of differentiation on chromosomes 1, 3 and 4 as seen between MTci5 and MTci2. However, two new peaks, a potential moxidectin-specific peak (not found in any other dataset) on chromosome 3 (peak range: 15–20 Mb) and a prominent peak on chromosome 4 (peak range: 85–88 Mb), also found in the between-farm analyses, and possibly the Choi comparison, were apparent. Notably, the main chromosome 5 peak found in almost all other ivermectin-resistant comparisons was absent in MTci7. However, a peak on chromosome 5 (peak range: 73–76 Mb), found in the MTci2:MTci5 comparisons and between-farm comparisons, was more prominent in the MTci2:MTci7 comparison than in the MTci2:MTci5 comparisons.

## Genetic variation associated with individual drug classes

Identifying common peaks of differentiation between the three experimental approaches allowed us to assign some peaks to specific drug classes. We have compiled a list of candidate genes associated with individual drug classes and previously described variants associated with resistance (**S7 Table**). We used these gene lists and peak positions to associate specific genes with resistance (summarised in **Table 2**).

**Benzimidazoles.** Almost all samples were phenotypically resistant to benzimidazoles to some degree, reflective of their widespread use over many years and the ease by which *T. circumcincta* can develop resistance to treatment. Only the $S_{inbred}$ isolate was fully susceptible, whereas MTci1 and MTci2 have each been previously suspected to harbour low levels of resistance [29], and the remaining isolates were considered to be resistant. Consistent among all comparisons that included a phenotypically well-defined resistant isolate with a phenotypically

**Table 2. Summary of the main genomic regions and candidate genes associated with each anthelmintic class.**

| Anthelmintic | Key regions[1] | Previously identified genes within loci[2] | Comparisons |
|---|---|---|---|
| Benzimidazoles | Chr 1: 60–65 Mb (260 genes in the region) | *beta-tubulin isotype 1* (TCIR_10026770; 62.287–62.292 Mb) | Between post-treatment farm samples Choi $S_{inbred}$:RS[3] Strains all MTci5:MTci2 comparisons (tallest in MTci5(BZ):MTci2), MTci7:MTci2 |
| | Chr 2: 60–67.5 Mb (329 genes in the region) | *beta-tubulin isotype 2* (TCIR_1005973; 59.147–59.153 Mb) | Choi $S_{inbred}$:RS[3] Between post-treatment farm samples and Farm 2 Pre:Post |
| Levamisole | Chr X: 1.5–5 Mb (130 genes in the region) | *acr-8* (TCIR_10183160; 1.830–1.857 Mb) *bar−1* (TCIR_00168840; 3.842–3.861 Mb) | Choi $S_{inbred}$:RS[3] |
| Ivermectin | Chr 5: 34–38 Mb (130 genes in the region) | *k07c11.10* (BBS10 ortholog) (TCIR_10160100; 35.385−35.390 Mb) *k07c11.10* (BBS10 ortholog) (TCIR_10160450; 35.954−35.956 Mb) | Farm Pre-Post comparisons Choi $S_{inbred}$:RS[3] Strains all MTci5:MTci2 (tallest peak in MTci5 (IVM):MTci2) |
| | Chr 5: 36–38 Mb (CMH) (61 genes in the region) | na | CMH: Farm1 Pre-Post + Farm2 Pre-Post |
| Moxidectin | Chr 3: 15–20 Mb (232 genes in the region) | *hlh−4* (TCIR_00074870; 17.657−17.661 Mb) | Strains MTci7:MTci2 |
| Multi-drug resistance/Other notable peaks | Chr 4: 65–72 Mb (251 genes in the region) | *osm−5* (TCIR_10130600; 66.113−66.126 Mb) *osm−3* (TCIR_00124060; 67.078−67.126 Mb) *haf−6* (TCIR_00124860; 69.059−69.065 Mb) *vab−2* (TCIR_10132650; 71.450−71.462 Mb) *vab−2* (TCIR_10132660; 71.510−71.520 Mb) | Between post-treatment farm samples Choi $S_{inbred}$:RS[3] Strains, all comparisons |
| | Chr 4: 85–88 Mb (158 genes in the region) | *pgp-9* (TCIR_00132080; 87.265–87.291 Mb) Putative ivermectin−related genes also within the peak : *vab−3* (TCIR_10139470; 85.910−85.932 Mb) *unc−44* (TCIR_00131600; 86.283−86.305 Mb) | Between post-treatment farm samples Choi $S_{inbred}$:RS[3] Strains MTci7:MTci2, perhaps also MTci5: MTci2 |
| | Chr 5: 73–76 Mb (150 genes in the region) | *daf−6* (TCIR_00160170; 75.740−75.769 Mb) | Between post-treatment farm samples Strains - most prominent in MTci7:MTci2 and least obvious in MTci5(IVM):MTci2 |

[1] Narrowest peak reported

[2] Note that genes in grey have been identified through *Caenorhabditis elegans* studies, but not yet in parasitic nematodes

sensitive isolate was a peak on chromosome 1 that contained *beta-tubulin isotype 1* (TCIR_10026770; gene position: 62.287–62.292 Mb), a gene well-established in its role in benzimidazole resistance. Three non-synonymous variants at positions P167, P198, and P200 of *beta-tubulin isotype 1* are broadly associated with resistance in several nematode species [30–32]; here, we identify the F200Y allele at high frequency in all strongly resistant isolates, as well as the E198L variant in the Choi (RS[3]), Farm and MTci5 data, albeit at a lower frequency (**Fig 3**). Consistent with Choi *et al.*, we did not find any haplotypes containing resistant alleles for both E198L and F200Y in any pooled sequencing dataset (**S8 Table**), further supporting the hypothesis that heterozygotes for both alleles are in trans- rather than cis- and that individuals homozygous for both resistance alleles are unviable [27].

In the predominantly susceptible MTci1 and MTci2 isolates, we see low levels of the F200Y variant, supporting the previous observations of suspected phenotypic resistance in these isolates and in agreement with the finding of low levels of the F200Y allele in MTci2 by Skuce and colleagues [29]. Additional *beta-tubulin* genes have been associated with resistance and can drive higher resistance levels than the *isotype 1* variants alone [16,33]. Five *beta-tubulin* genes were identified in total in the tci2_wsi3.0 assembly, although not all have been associated with resistance previously. Here, we find the E198A variant in the *beta-tubulin isotype 2* gene

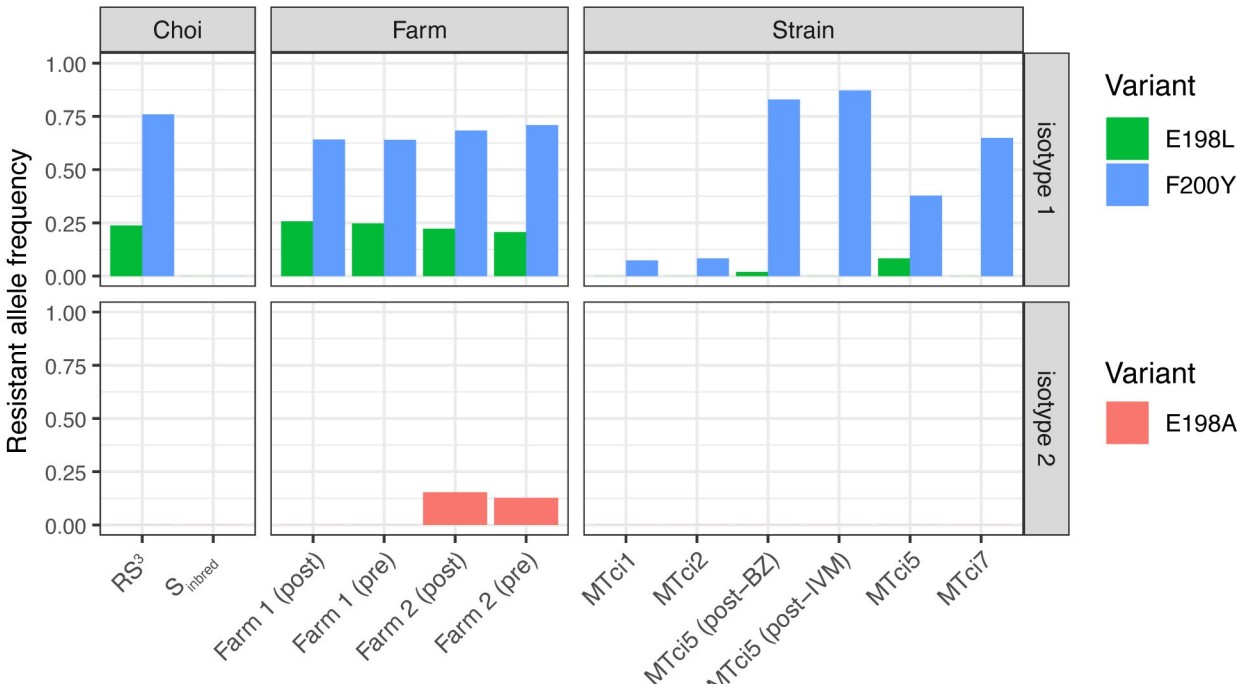

**Fig 3. Characteristics of genetic variation associated with resistance to benzimidazole treatment.** Resistance to benzimidazole treatment is strongly associated with variation in the *beta-tubulin isotype 1* gene (TCIR_10026770). In most samples, high to moderate variant allele frequencies from whole genome sequencing data were observed for resistance variants F200Y and E198L, respectively. The $S_{inbred}$ was the only wholly susceptible sample, whereas "susceptible" MTci1 and MTci2 have low levels of resistant variant F200Y. Resistance to higher levels of benzimidazole treatment has previously been associated with variation in the *beta-tubulin isotype 2* gene (TCIR_10059730); here, we find the E198A variant only in the Farm 2 isolate.

(TCIR_10059730) in the Farm 2 isolate only (**Fig 3**; $freq_{E198A}$ = 12.7% and 15.4% in the pre and post-ivermectin-treatment samples, respectively). The presence of E198A in Farm 2 but not Farm 1 may explain the difference in benzimidazole treatment efficacy between the two farms ($FECR_{benzimidazole}$ = 87.7% and 4.5% for Farm 1 and Farm 2, respectively), despite both having a very similar frequency of *beta-tubulin isotype 1* resistant variants at both positions E198L and F200Y (**Fig 3**).

**Levamisole.**    Three isolates were phenotypically resistant to levamisole–RS[3], MTci5 and MTci7 –however, only RS[3] was directly exposed to levamisole immediately before sample collection. Despite the phenotypic history of the isolates, no shared peaks of genetic differentiation among the resistant isolates were identified that could be attributed to levamisole resistance. A single peak on the X chromosome unique to the $S_{inbred}$:RS[3] comparison contained the acetylcholine receptor *acr-8* (TCIR_10183160) (peak range: 1.5–5 Mb; gene position: 1.830–1.857 Mb), previously identified as associated with levamisole resistance (**Fig 1B**). In *H. contortus*, the S168T variant of *acr-8* has been strongly implicated as being causal, having been functionally validated [34] and found in several levamisole-resistant populations around the world [16,35,36]; here, we identify the homologous variant S167T found at high frequency in RS[3] (tci2_wsi3.0_chr_X, pos: 1,846,323, $freq_{S167T}$ = 96.4%). This variant was, however, not found in the remaining levamisole-resistant isolates; the only other sample to contain it in a single sequencing read was Farm 2 ($freq_{S167T}$ = 2%). The absence of the *acr-8*-associated peak in MTci5 and MTci7 suggests that other genetic variants may drive resistance; an alternative explanation, based on extensive experience with the cryopreservation of isolates, is that sample

storage in liquid nitrogen can reduce the levels of levamisole resistance of an isolate post-recovery. We note that the levamisole resistance phenotype of the individual samples used was unknown. A key challenge to identifying these regions is that both isolates are also resistant to ivermectin and benzimidazoles, confounding the identification of drug-specific peaks. A review of previously described candidate genes associated with levamisole resistance highlighted this challenge (**S7 Table**); for example, the acetylcholine receptor subunit *unc-29* (TCIR_10027440) was found to be closely located alongside *beta-tubulin isotype 1* on chromosome 1 and, therefore, in these studies, if a genetic signal of differentiation associated with *unc-29* were present, it would not be distinguished from the established signal associated with benzimidazole resistance. To do so, recombination mapping to identify independently segregating variants using sequencing of single-worms, phenotyped for levamisole sensitivity, could help map physically close variants associated with different drug classes. However, other subunits such as *unc-63* (TCIR_10007740) on chromosome 1 at ~15.47 Mb and the levamisole-receptor genes *lev-1.1* (TCIR_10117820) and *lev-1.2* (TCIR_10117830) located on chromosome 4 at ~31.74 and ~31.81 Mb showed no clear genetic association with resistance.

**Moxidectin.** Only the MTci7 isolate was moxidectin-resistant, so it is difficult to make robust conclusions from the analysis of a single isolate. Moxidectin is a macrocyclic lactone and belongs to the same drug class as ivermectin. Phenotypically, resistance to moxidectin typically confers resistance to ivermectin; however, resistance to moxidectin may occur independently of ivermectin resistance in *C. elegans* [37], a phenomenon also found in a recent field study of gastrointestinal nematodes on sheep farms [38]. More commonly, however, moxidectin can still be used to control nematode parasites in ruminants following a diagnosis of ivermectin resistance for a period of time [39], which may be due either to increased potency (a higher lipophilicity) or a different mechanism/inheritance of resistance [40]. A single peak on chromosome 3 (peak range: 15–20 Mb) found only in the MTci7:MTci2 comparison and not in the other ivermectin-resistant isolate comparisons is suggestive of a moxidectin-specific response. This peak contains the ortholog of *hlh-4* (TCIR_00074870), the mutation of which has resulted in ivermectin and moxidectin resistance in *C. elegans* [37]. In *C. elegans*, *hlh-4* is a transcription factor located within ADL neuronal cells that determines amphid neuron cell fate and influences the expression of a range of genes associated with ADL cells, including neurotransmitters, neuropeptides, ion channels, and electrical synapse proteins [41].

Moxidectin and ivermectin may share molecular targets, however, the genetic response to selection by each drug on those targets may differ. This could, in part, explain some differences in MTci2 comparisons with MTci7 relative to ivermectin-resistant isolates, such as the absence of the ivermectin peak on chromosome 5 or the more prominent peaks on chromosomes 4 (peak range: 85–88 Mb) and 5 (peak range: 73–76 Mb) in MTci2:MTci7 relative to MTci2:MTci5.

**Ivermectin.** All comparisons involving ivermectin-resistant isolates, except for MTci7, revealed a common peak on chromosome 5 (consensus peak range: ~35–40 Mb). This peak was the only peak identified in the pre-to-post ivermectin treatment comparisons of the two farm populations, suggesting it was specifically an ivermectin-treatment-related response. To explore this region further, we focused on the Farm 1 population, in which we compared $F$st from pooled sequencing (**Fig 4A**) with an independent experiment using ddRAD-seq to assess linkage disequilibrium (LD) among genotypes from individually re-sequenced larvae from the same pre- and post-treatment sample populations (**Fig 4B**). These analyses complemented each other, showing a region of differentiation and increased linkage in the post-treatment population at approximately 35–38 Mb, consistent with a reduction in diversity associated with drug-mediated selection. Two other peaks of LD were evident on chromosome 5, one of which corresponded with an $F$st peak at approximately 73–76 Mb in the genome-wide Farm 1 vs Farm 2 post-treatment comparison and the MTci2 comparisons with MTci5 or MTci7

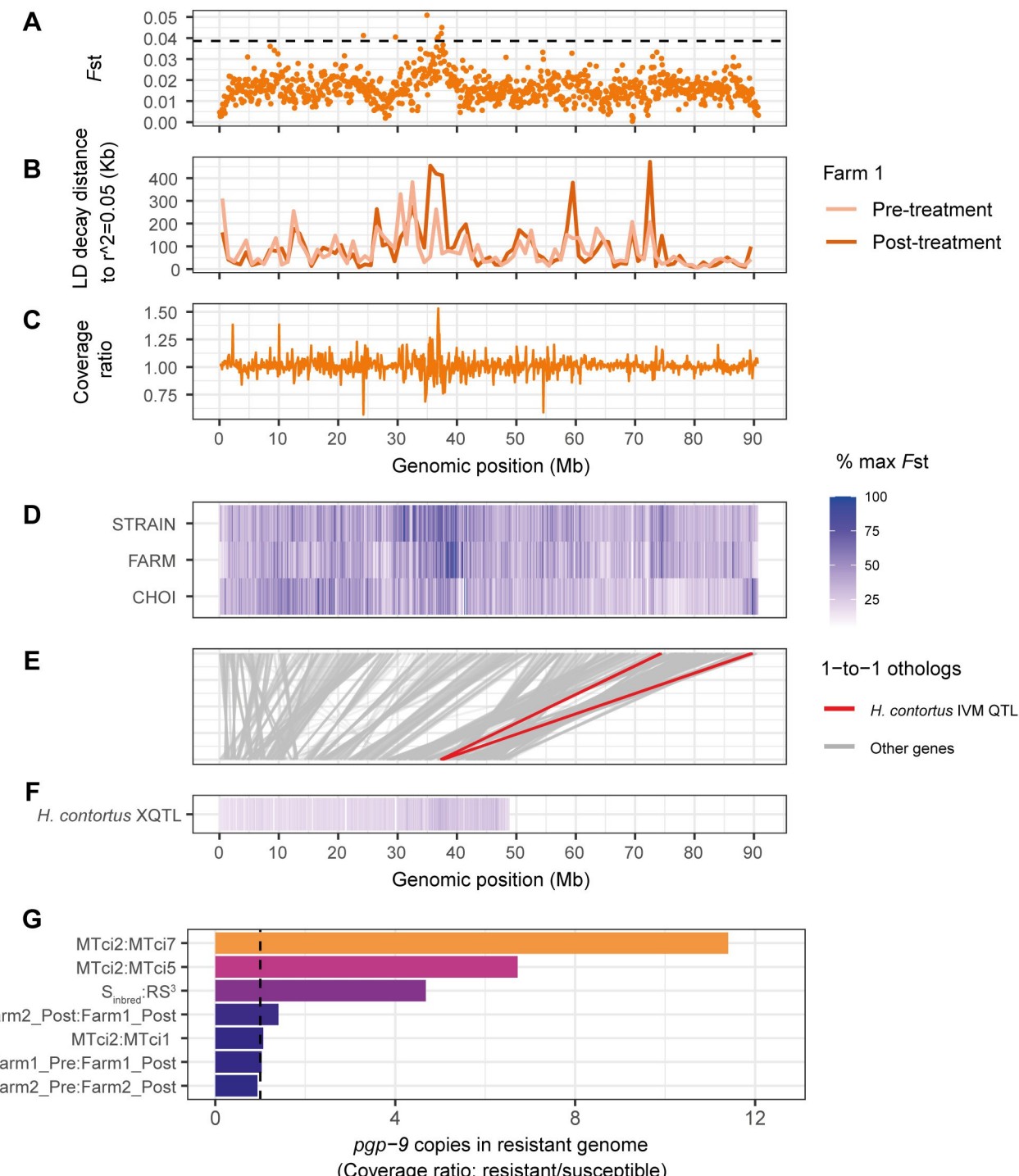

**Fig 4. Characteristics of genetic variation associated with ivermectin resistance. A, B, C.** Analysis of the major peak of differentiation on chromosome 5, comparing *F*st between Farm 1 pre- and post-treatment pooled sequencing samples in 100 kb windows (A), linkage disequilibrium from ddRAD-seq of individual larvae from Farm 1 pre- and post-treatment populations (B), and relative sequencing coverage ratio between Farm 1 pre- and post-treatment pooled sequencing samples in 100 kb windows. **D.E.F.** Comparison of shared and unique signals genetic differentiation associated with ivermectin response on chromosome 5 of *Teladorsagia circumcincta* (D) and *Haemonchus contortus* (F). In both the D and F panels, data is coloured by the maximum *F*st value in each experiment (low = white, high = blue). The middle panel (E) shows the relative position of 1-to-1 orthologous genes shared between the two species (n = 1,253 genes on chromosome 5, indicated by grey lines). The orthologous genes in the major QTL associated with ivermectin response in *H. contortus* are highlighted in red. **G.** Estimation of *pgp-9* (TCIR_00132080) copy number expansion in resistant relative to susceptible isolates. The vertical dashed line is positioned at a copy number of one, representing the expected copy number ratio between ivermectin-susceptible isolates.

(**Fig 2A and 2C**). To further explore this peak of differentiation at 35–38 Mb, we assessed coverage (**Figs 4C** and **Fig F in S1 Appendix**), as well as gene, BUSCO, and repeat density in this region (**Fig G in S1 Appendix**); while there was increased variance in coverage in discrete regions of the locus, it did not explain the entire signal of differentiation, and there was no apparent bias in gene or repeat density suggesting it was an artefact of the assembly. Interestingly, variation in coverage was noted only between ~35–38 Mb between pre- and post-ivermectin samples on Farm 1, while the variation extended to 40 Mb on Farm 2, and, when post-ivermectin samples were compared (F2A:F1), the variation in coverage was primarily between 38–40 Mb, corresponding to the peak of genetic differentiation between the farms. These results collectively suggest that the locus under ivermectin selection lies between 35–38 Mb, although due to the lower coverage of RADloci between 34 and 35 Mb, this region cannot be excluded. To search for SNP alleles under selection by ivermectin on both farms, a Cochran-Mantel-Haenszel test was performed. This accentuated the significance of the 36–38 Mb region in particular (**Fig H in S1 Appendix**). Two SNPs were identified (Benjamini-Hochberg FDR 5%) as being significant following correction for multiple testing; Chr 5:38,037,930, peripheral to the ivermectin locus as defined here, and Chr 5:36,505,260, predicted to be an intron variant of TCIR_00147810, an orthologue of a *H. contortus* G protein receptor domain containing protein. However, given the availability of only two pairwise comparisons for the CMH test, additional experiments and replicates are needed to confirm whether these particular SNPs are important in ivermectin resistance, and whether multiple variants may individually (as for benzimidazoles or monepantel), or collectively confer ivermectin resistance. In our more conservative estimate of the ivermectin peak, ranging from 34–38 Mb, a total of 130 genes were identified in the region (**S8 Table**); considering that no strongly supported candidate genes were found, these results collectively lead us to conclude that this region has a novel driver of resistance.

The shared peak between ivermectin-resistant isolates on chromosome 5 prompted us to compare it with the orthologous chromosome 5 of *H. contortus*. In previous works on *H. contortus*, we found a single peak associated with ivermectin resistance located at 37.5 Mb on chromosome 5, from which we proposed that overexpression of the bHLH transcription factor *cky-1* was associated with resistance [16,42]. Comparison of *F*st data highlighting the regions of differentiation between susceptible and resistant isolates of both *T. circumcincta* (**Fig 4D**) and *H. contortus* (**Fig 4F**), plotted together with 1-to-1 orthologs between the chromosomes of the two species (**Fig 4E**; grey lines), showed that the two major peaks in each species did not share genes and, therefore, are interpreted to be unrelated. However, a comparison of the orthologous genes within the *H. contortus* extreme quantitative trait locus (XQTL) peak with the *T. circumcincta* genome reveals two regions of interest (**Fig 4E**; red lines), with minor but significant peaks of differentiation between susceptible and resistant *T. circumcincta*. One of these peaks–found only in the Choi data–contains the *T. circumcincta* ortholog of *cky-1* (TCIR_10182250). Considering that our previous hypothesis of *cky-1* overexpression led to resistance in *H. contortus*, we had previously tested and confirmed that *cky-1* was significantly higher expressed in the ivermectin-resistant MTci5 relative to the susceptible MTci2 isolate (see **Fig 5E** of [16]). We do not, however, see evidence of a peak of genetic differentiation between MTci5 and MTci2 around *cky-1*; this observation does not invalidate the previous hypothesis (i.e., that *cky-1* is a crucial mediator of ivermectin resistance) but would require that *cky-1* expression is influenced by selection in another region of the genome for it also to be involved in ivermectin resistance in MTci5. Alternative hypotheses include (i) that the two peaks in *T. circumcincta* associated with the single *H. contortus* peak represent a misassembly in the *T. circumcincta* genome and, if corrected, would resolve into a single peak in each, or (ii) that *cky-1* at the end of the chromosome belongs in the prominent peak shared by all

ivermectin resistant *T. circumcincta* isolates; however, careful examination of the Hi-C data, which should be sensitive to detecting misassemblies at this scale, does not support these alternate hypotheses.

The peak on chromosome 4 at ~85–88 Mb–prominent in the MTci2:MTci7 comparison and between-farm post-treatment populations–contains the ortholog of the p-glycoprotein gene *pgp-9* (TCIR_00132080, 4:87.265–87.291 Mb). P-glycoproteins are members of the ATP-binding cassette (ABC) transporters, which function at the cellular membrane to pump foreign molecules out of the cell and are broadly implicated in resistance to chemotherapeutic drugs, including anthelmintics [43,44]. As such, the family of P-glycoproteins have been proposed to play a role in ivermectin resistance in several nematode species [45–47]. In *T. circumcincta*, constitutive over-expression of *pgp-9* in triple-resistant MTci5 [48] and increased genomic copies of *pgp-9* in the RS$^3$ isolate relative to the S$_{inbred}$ draft genome assembly and by qPCR of triple-selected adult males [27] have been described, both of which are consistent with positive selection to increase the rate by which anthelmintics, and possibly ivermectin in particular, are removed from cells. To assess *pgp-9* copy number variation, we calculated the relative gene coverage genome-wide between susceptible and resistant isolates (**Fig I in S1 Appendix**), revealing *pgp-9* as an outlier in multi-drug-resistant isolates. Comparatively, *pgp-9* copies were highest in the resistant isolates in the following comparisons: MTci2:MTci7, followed by MTci2:MTci5 and RS$^3$:Sinbred (**Fig 4G**), consistent with previous studies. The relative height of the *pgp-9* peak to other peaks associated with ivermectin response is suggestive that *pgp-9* variation is not the primary driver of ivermectin resistance in MTci5 and RS$^3$ isolates but could be a strong candidate in MTci7. However, this cannot be attributed explicitly to ivermectin resistance in this study.

Additional smaller peaks of differentiation were identified between MTci2:MTci5, some of which contained genes previously associated with ivermectin resistance in *C. elegans*. For example, a gene associated with amphid channel morphology mutants resulting in ivermectin and moxidectin resistance in *C. elegans*, *daf-6*, is the ortholog of TCIR_00160170, found within the peak ~75 Mb along chromosome 5. It is unknown if variation in this gene, or others in minor peaks, are associated with resistant phenotypes in *T. circumcincta*, but warrant further investigation.

## Discussion

The chromosomal assembly and curated annotation of the MTci2 isolate of *T. circumcincta* provide an important addition to the growing number of high-quality resources for clade V nematodes and comparator for several important human and animal pathogens. The assembly, now with chromosome-scale scaffolds consistent with its karyotype [49] and that of closely related clade V species, is significantly more complete in both genome and annotation content than existing *T. circumcincta* assemblies [27,28], facilitated by long-read and long-molecule sequencing and novel approaches such as using RNA-seq and DNA-seq data for genome polishing. We improved the annotations further using manual curation; over 3,200 gene models were updated (~14% of all predicted genes), with an initial focus on genes encoding ion channels, vaccine candidates, and those previously associated with anthelmintic resistance. We could not fully assign all contig sequences to chromosomes, choosing instead to keep any with at least partial unique sequences as either unplaced scaffolds or assigned to linkage groups. The rapid advances in low-input sequencing (which now requires orders of magnitude less starting material than when this genome project was initiated) [50,51] suggest that further improvements could be achieved with investment in single worm genome long-read sequencing, or with ultra-long sequencing reads such as those obtained from Nanopore single molecule sequencing for assembly or further scaffolding.

Our primary motivation for improving the *T. circumcincta* genome assembly was to provide a resource to help resolve the genetics of drug resistance. Interpreting genetic signals associated with drug resistance using fragmented genome assemblies is a significant technical challenge and can easily lead to misinterpretation. For example, recent genetic comparisons of the same data from susceptible and resistant isolates mapped to three different versions of the *H. contortus* genome could have led to entirely different interpretations of the number of genes and mechanisms involved in resistance (see Fig 7 of [52]). Here, we have used three experimental approaches to provide insights into selection in response to treatment with different anthelmintic classes.

Our analyses of pre-and-post ivermectin selection on two UK farms revealed a common signature of ivermectin selection on chromosome 5, which we propose contains a novel driver of resistance. No other regions appeared obviously under selection by ivermectin. However, a comparison of the post-treatment groups revealed a more complex history of selection by anthelmintics. Consistent with the fact that both farms had high levels of phenotypic benzimidazole resistance (by FECRT), a strong genetic signal was identified around the *beta-tubulin isotype 1* locus. Mutations previously associated with benzimidazole resistance were also identified on each farm. Furthermore, the between-farm differentiation at this locus and adjacent to the ivermectin-treatment-associated locus (from 37.5–40 Mb on chromosome 5) likely reflects that selection has occurred independently on different genetic backgrounds rather than appearing on one farm and spreading to others, consistent with previous work [12]. The use of ivermectin for the treatment of ovine helminths is typically intercurrent with other anthelmintic classes, for example, as a subsequent response to the lack of efficacy of benzimidazoles due to rotation between drug classes or as part of seasonal targeting of parasites. In the UK, for instance, it is common to treat lambs to control *Nematodirus battus* infections with benzimidazoles in spring but then ivermectin later in the grazing season, based on perceived effectiveness [53]. In some parts of the world, combination products containing more than one nematode anthelmintic are used. Therefore, globally, ivermectin resistance commonly appears on a background of benzimidazole resistance, and while the mechanisms of action are distinct between these two drug classes, these co-occurring signals of selection can confound the interpretation of resistance mechanisms. Careful study design can help to pull these co-occurring signals apart. For example, the use of a single anthelmintic and comparison of pre and post-treatment samples from an admixed and multi-drug resistant population, as shown here, could help, or via the use of genetic crosses as discussed below. Further phenotyping of sample populations *in vitro*, with subsequent individual or pooled sequencing, may also serve to distinguish selection signals. Importantly, these experiments will only be likely to work if the parasite population has sub-populations that are sensitive to one of the anthelmintics (as shown here), as otherwise, if the population is made up of only dual-resistant individuals (no signal will be seen) or either dual-sensitive or dual-resistant individuals (both signals will be seen), the experiment is less likely to be successful. Understanding the genetic relatedness of haplotypes in these regions, particularly from a denser sampling of parasite populations, will further inform how resistance evolves and spreads.

Genetic crosses followed by drug selection have been important for refining our understanding of resistance evolution in several helminth species [21,23,27,52,54,55]. Our reanalysis of genomic data from drug-susceptible and multi-drug selected, backcrossed *T. circumcincta* isolates from New Zealand [27]–from which the authors of the original analyses concluded that multi-drug resistance is a polygenic, quantitative trait–specifically addressed the hypothesis that an improved genome assembly would increase the resolution of selection signals. This hypothesis was partly supported, especially when the Choi data was interpreted in the context of the farm and UK strain datasets, revealing some signals of selection around known

candidate genes, for example, *beta-tubulin isotype 1*, *acr-8* and *cky-1*. However, these and other genomic regions were difficult to interpret using the data from Choi and colleagues independently. We interpret these more complex signals as being caused by the experimental framework of the original genetic cross, in which an intense co-selection with three anthelmintics—benzimidazole, levamisole, and ivermectin—was used. Considering the genes associated with resistance to these anthelmintics are located on different chromosomes, co-selection would likely have resulted in severe bottlenecks in the selected population, reducing the potential for recombination needed to break down the linkage between causal and background genetic variation, and likely explains the broader peak regions of genetic differentiation around candidate genes, i.e., more than half of chromosome 1 is differentiated between $RS^3$ and $S_{inbred}$ around the *beta-tubulin isotype-1* locus. A similar cross framework followed by drug selection with individual drug classes would better refine the signals to reveal drug-specific peaks.

We identified broadly consistent signals of selection between geographically distinct populations of the UK and New Zealand, reflecting independently acquired but conserved pathways to resistance. Additional loci associated with resistance are likely to exist within these data but are difficult to confidently identify using sequencing data from pools of individuals; similarly, it will be important to determine the relevance of these signatures of selection in other drug-resistant *T. circumcincta* from around the world. However, it is clear that having a contiguous genome assembly and whole genome sequencing alone is not sufficient to resolve the genetics of resistance in *T. circumcincta*. Moreover, the genetics of ivermectin resistance in *T. circumcincta* may be different than in the closely related *H. contortus*, suggesting that the genetics of ivermectin resistance may be more complex in strongyle nematodes than benzimidazole or monepantel resistance. For example, ivermectin resistance in *H. contortus* is consistently associated with a major QTL located on chromosome 5 in globally distributed resistant populations [5,18,52] and is proposed to be driven by selection to over-express the transcription factor *cky-1* [16,42]. However, while over-expression of *cky-1* has been shown previously in ivermectin-resistant *T. circumcincta*, in this study, evidence of selection in the genome was inconsistent and only observed between the Choi isolates. It is possible that *cky-1* is not the direct target here, but that selection still acts on a shared molecular pathway, which remains unknown. We acknowledge that the experimental frameworks of controlled genetic crosses and drug selection in *H. contortus* to identify precise drug-associated genetic variation and sequencing field isolates of *T. circumcincta* here are sufficiently different and, therefore, will have different resolutions. Equally, unlike benzimidazole resistance, the genetic mechanisms that lead to ivermectin resistance may vary between species. Nonetheless, using three different experimental frameworks and comparative approaches applied here provides insights into the evolutionary processes by which resistance develops. For example, the distinction of *beta-tubulin* and ivermectin-associated peaks between the post-treatment farm populations suggests that resistance has evolved independently on diverse haplotypes instead of evolving once followed by radiation. This has implications for diagnostics; where the same SNP is involved across species, allele-specific techniques can be used to diagnose and monitor resistant variants [56]. Even where several closely spaced SNPs may independently confer resistance, as in the case of the three SNPs in the *beta-tubulin isotype 1* gene, DNA sequencing approaches such as amplicon sequencing can identify them [57]. For monepantel resistance, a variety of mutations, each independently likely to alter the function of *mptl-1*, have been identified in phenotypically resistant *T. circumcincta* populations [58], but the diagnostic utility of this information is less clear and may require an understanding of haplotypes of variants across a larger region; such variants might be monitored by long-read approaches such as PacBio or Nanopore sequencing. Nonetheless, in the research setting, the ability to monitor resistant genotypes in specific species via sequencing technologies will allow a more detailed approach

to anthelmintic resistance risk and refugia studies [59], given the limitations of the FECRT, which necessitates the use of anthelmintics. Even without a causal mutation being identified, the ability to monitor for reduction in diversity and changes in haplotypes over a small region of the genome will enable the study of more complex loci and resistance mechanisms.

There were limitations to this study. It is still challenging to assemble genomes of genetically diverse pools of organisms; here, pools of thousands of larvae were required to obtain sufficient DNA for PacBio sequencing, resulting in haplotypic diversity and artefactual duplication that still exists in the assembly and needs to be resolved. Sequencing of relatively small pools of resistant and susceptible populations to compare genome-wide variation provides a limited level of resolution, further challenged by using broad 100 kb windows to maximise the signals of differentiation associated with resistance from stochastic noise; this could be refined further by analysing genotypes from large-scale single worm sequencing of phenotypically well-defined populations, albeit at a significantly greater cost than pooled sequencing to measure allele frequencies. Finally, comparing pre and post-ivermectin treatment parasite populations has lowered the background genetic variation commonly observed for genetic comparisons of distinct isolates, allowing the identification of a single locus of differentiation. In contrast, between isolate comparisons identified multiple peaks of differentiation which could not be confidently assigned to specific drug classes and may reflect non-drug-related selection and divergence between populations. Treatment directly before sampling a population was found to enhance the signal of anthelmintic-specific selection in the UK MTci5 isolate, but even so, these loci were not the only differentiated regions with MTci2 and were primarily interpreted in relation to the wider data. This could be resolved through genetic crosses followed by selection with individual drug classes to focus on drug-specific responses. Although working with highly heterozygous populations, which are sexually reproducing obligate parasites and which yield very low quantities of DNA per worm (e.g. < 1 ng per L3, the most accessible free-living stage), will always be a challenge, the reducing cost of amplicon sequencing and improved methods for sequencing ultra-low input DNA [50,60] provides scope to perform the sequencing of many more individuals than were used in the ddRAD-Seq experiment here. Such work may assist in narrowing our anthelmintic resistance loci. The new genome assembly also allows for older techniques, such as microsatellites, which were successful in narrowing down ivermectin resistant to chromosome 5 in *H. contortus* [54] and an artemisinin resistance marker on chromosome 13 of *Plasmodium falciparum* [61], to be reconsidered, as the relative position of each microsatellite within the genome can now be known. These may provide a more universal scope for investigating resistance where sequencing resources or funds are limited.

In summary, the MTci2 chromosomal assembly and curated annotation provide a robust resource for genome-wide comparisons of *T. circumcincta*. We have identified regions of the genome associated with distinct anthelmintic classes, prioritising some known genes and highlighting previously uncharacterised genes associated with resistance. Targeted, fine-scale mapping using samples from drug-specific experiments will likely yield further insight into the specific causal variants driving resistance, opening new avenues toward using genetics to inform more effective and sustainable control of helminths that pose substantial global animal health and economic problems.

## Methods

### Parasite material for the genome assembly

The genome assembly described was generated from the *T. circumcincta* "Weybridge" isolate, MTci2. The isolate was originally collected pre-1990 by the Central Veterinary Laboratories,

Weybridge, UK, and has been maintained at the Moredun Research Institute (MRI) by serial passage in sheep since 2000. MTci2 is susceptible to most anthelmintic classes; however, low levels of benzimidazole resistance have been suspected [29].

## Genome assembly

**PacBio long-read sequencing.**   Two PacBio long-read datasets were generated. The first long-read data were derived from whole genome amplified (WGA) DNA (REPLI-g Single Cell Kit, Qiagen) from two single adult male worms; the rationale of using WGA was to obtain micrograms of DNA for library preparation from the low amount of starting material (~10 ng per worm) while maintaining low genetic diversity from outbred samples. SMRTbell sequencing libraries prepared following the '20 kb Template Preparation Using BluePippin Size-selection System' protocol were sequenced using 61 SMRT cells on the PacBio RSII platform, generating 6.5 million reads (N50 = 6.8 kb) representing ~63X coverage of the estimated 700 Mb genome (**S1 Table**). Initial assemblies using these data were highly fragmented, which we attributed to a high frequency of chimeric sequences in the WGA data. To address this, we made sequencing libraries from DNA obtained from a pool of 250,000 larvae, which were sequenced using 15 SMRT cells on the PacBio Sequel platform. In total, 7.4 million reads (N50 = 11 kb) were generated, representing ~74X raw coverage. Although these data contained significantly higher genetic diversity due to the large number of pooled larvae used, the issues with chimeric reads were resolved as WGA was not used.

**Illumina Hi-C library preparation and sequencing.**   Hi-C libraries were prepared using the Arima-Hi-C Kit for Animal Tissues (Arima Genomics) following the manufacturer's instructions with 20 frozen adult female worms as input. An Illumina sequencing library was made using the Accel-NGS 2S Plus DNA Library Kit (Swift Biosciences) with a modified Arima Genomics protocol for low-input material. The library was sequenced on a single lane of Illumina HiSeq X10 using 150 bp paired-end chemistry, generating >150x coverage of the genome (**S1 Table**).

**Genome assembly.**   The PacBio Sequel data were initially assembled using *Canu* v.1.9 [62] (parameters: corMhapSensitivity = high corMinCoverage = 0). The high genetic diversity within the pooled larvae resulted in a highly haplotypic (BUSCO: 66% duplicated) and expanded assembly size (1.57 Gb), approximately twice as large as previous estimates (700.8 Mb; [27]). To reduce redundant haplotypes, we used *purge_dups* v.1.2.5 [63] to deduplicate the assembly, followed by *Redundans* v.0.14a [64] to re-scaffold using long reads after haplotypic sequences had been removed. *Redundans* and *purge_dups* were assessed separately and together on their ability to improve contiguity and decrease assembly size by eliminating redundant sequences while maximising complete and minimising duplicated BUSCOs.

Further assembly scaffolding was performed using the Illumina Hi-C data and *Salsa2* [65]. Two rounds of scaffolding with *Salsa2* were performed, with manual assembly curation between and after scaffolding. Manual curation was performed using *pretext* (https://github.com/wtsi-hpag/PretextMap; https://github.com/wtsi-hpag/PretextView) with *HiGlass* [66] contact maps as a guide to identifying misassembled regions and placing unassigned contigs and scaffolds into their chromosomal context. Where gaps in the assembly were created during curation, padding of 10 Ns were added. The Arima mapping pipeline (https://github.com/ArimaGenomics/mapping_pipeline) was used to generate data for curation, except that *minimap2* rather than *bwa-mem* was used for mapping the Hi-C reads. Additional supporting analyses, including the calculation of PacBio read coverage (*minimap2* -x map-pb), positions of repeats (*Red* [67]), assembly gaps (*seqtk*; https://github.com/lh3/seqtk), and telomeres (searching for a dimer of GCCTAA), were used to guide curation. The reference-aware scaffolder,

*RagTag* v.2.1.0 [68], was used to assign the near-chromosome scaffolds into five autosomes and an X chromosome based on the chromosomal assembly of the closely related parasite *H. contortus* [69] after the first round of manual curation.

The assembly was further curated based on gene and sequence content. Before using Hi-C data, we observed that our original draft assembly contained single-copy BUSCO sequences that were missing from the Hi-C scaffolded assembly. Using an iterative and comparative approach, we recovered sequences from the draft assembly containing these missing BUSCO genes, merged them with the chromosomal assembly, and then removed redundant, contained contig and scaffold sequences (i.e., sequences that map within the boundaries of another sequence) by comparison using *nucmer* v4.0.0 [70]. This approach was followed with *Blobtools* [71] to remove potential contaminants, retaining sequences classified as 'Nematoda' by blast and unclassified sequences. Following this, the Arima mapping pipeline was repeated, and a final round of manual curation using *Pretext* and *HiGlass* was performed.

Finally, gaps were identified and filled using *TGS-GapCloser* v.1.2.1 [72], followed by polishing the assembly with long PacBio reads using *arrow* v.2.3.3 and three rounds of polishing using *POLCA* [73], the first using the Hi-C whole-genome sequencing reads and the second using RNA-seq reads (see below), followed by a final round using the Hi-C genomic reads. The rationale for using RNA-seq reads was to correct potential indels in the coding sequences. However, although successful, as a side effect, this approach caused errors in splice donor and acceptor sequences, leading to poor annotations. These were corrected (as was the annotation) with the final round of polishing with genomic reads.

## Genome annotation

**Parasite material for genome annotation.** To support the genome annotation, we prepared RNA from L3, exsheathed L3, L4 (male and female), and adult (male and female) *T. circumcincta* MTci2. Briefly, L3 were obtained from coproculture from worm-free donor sheep faecal material containing eggs after incubation at 20˚C for 10–14 days. Exsheathment was performed as previously described [74]. L4 and adult stages were harvested seven and 28 days post-infection, respectively, as previously described [75]. All samples were washed three times in PBS and subsequently snap-frozen in liquid nitrogen before storage at -70˚C until RNA extraction.

**RNA extraction.** Total RNA was extracted using a Quick-RNA tissue/insect Microprep kit (Zymo Research). Worm samples were homogenised in 800 µl of RNA lysis buffer by bead beating in ZR BashingBead Lysis Tubes (0.1 & 2.0 mm) using a Precellys 24 tissue homogeniser (Bertin Instruments) (x3 30 sec pulses at 5000 rpm). Total RNA was purified according to the manufacturer's recommendations, with an on-column DNAaseI digest, and eluted in 15 µl nuclease-free water. RNA quantity and integrity were analysed using a NanoDrop One instrument (Thermo Fisher Scientific) and Agilent Bioanalyser 2100 system using RNA 6000 Nano kit. All RNA samples had a ribosome integrity number (RIN) $\geq$ 7. Total RNA was stored at -70˚C before further use.

**RNA-seq library preparation and sequencing.** All library synthesis and sequencing were performed by BGI Tech Solutions (Hong Kong). In brief, dual-indexed, strand-specific RNA-seq libraries were constructed from the submitted total RNA samples. The barcoded individual libraries were pooled and sequenced on a DNBseq G400 platform (paired-end, 2x150 bp sequencing), generating an average of 35.4 million reads per sample.

**Iso-seq library preparation and sequencing.** Total RNA was isolated from *T. circumcincta* xL3, L4, adult male and adult female, as described above. Library preparation and

sequencing were conducted at the University of Liverpool, Centre for Genomic Research. Samples were processed using a Teloprime full-length cDNA amplification kit v2 (Lexogen GmbH), following the manufacturer's protocol. This involved first-strand cDNA synthesis by denaturing samples (1000 ng) at 70˚C with primer RTP and, after cooling to 37˚C, mixing with reverse transcriptase and incubating at 46˚C for 50 min. Each sample was cleaned using the supplied columns, then ligated to a cap-dependent linker at 25˚C for 3 h and cleaned with the same columns. The samples were incubated with second strand synthesis mix for 90 s at 98˚C, 60 s at 62˚C, and 72˚C for 5 min, with a hold at 10˚C before being column-cleaned again. Half of each sample was amplified for 15 cycles (one cycle of 95.8˚C 30 s, 50˚C 45 s, 72˚C 20 min, followed by 14 cycles of 95.8˚C 30 s, 62˚C 30 s, 72˚C 20 min) and cleaned with a column as before. The amplified cDNA products were made into SMRTbell template libraries according to the Iso-Seq protocol by Pacific Biosciences. Sequencing was performed on the PacBio Sequel System, and 1 SMRT Cell was run for each sample with a movie run-time of 600 min for each SMRT Cell.

**Annotation.** The genome assembly was annotated using *BRAKER3* [76]. Mapped *T. circumcincta* RNA-seq reads, and *H. contortus* protein sequences were provided to *BRAKER3* as evidence. RNA-seq reads were mapped to the genome using *HiSat2* (with—dta flag; [77]) followed by name sorting using samtools. Predicted *H. contortus* protein sequences were obtained from WormBase ParaSite release 18 PRJEB506; these protein sequences were chosen due to the relatively close phylogenetic relationship between *T. circumcincta* and *H. contortus* and because the *H. contortus* annotation has been extensively manually curated [15].

The initial exploration of the output of *BRAKER3* revealed a large proportion of high-quality annotations, but some genes were not annotated by this pipeline despite the apparent support of RNA-seq mapping. We found that the *BRAKER3* intermediate files produced by *AUGUSTUS* had greater sensitivity to detect these missing annotations but with lower specificity. To recover these genes, we used *bedtools subtract* to collect the missing genes from the *AUGUSTUS* annotation, after which *gffread* was used to remove single exon transcripts. This approach resulted in a substantial increase in BUSCO scores after merging the *BRAKER3* and recovered *AUGUSTUS* features.

**Manual curation.** We used the genome annotation to further curate the assembly. Coding sequences annotated on linkage groups and unplaced scaffolds were aligned to the chromosomal scaffolds using *exonerate* (—model protein2genome—percent 50) [78]. Scaffolds and annotations that were syntenic with chromosomal scaffolds were subsequently removed from the assembly. This approach was further supported by removing scaffolds containing duplicated BUSCO sequences between scaffolds and chromosomes. Finally, manual curation of genes of interest (for example, ion channels and genes associated with resistance to at least one anthelmintic or those involved in vaccine development) and some surrounding regions was performed using *Apollo* 2 [79]. Using RNA-seq and Iso-seq mapping tracks as evidence, approximately 3,200 genes were edited in the first instance; however, this is an ongoing effort for which annotation updates will be made. These curated annotations were reintroduced into the *BRAKER3+AUGUSTUS* annotation described above after removing redundant and erroneous gene models using *bedtools subtract* and *AGAT* [80]. *AGAT* was further used to add start and stop annotations (*agat_sp_add_start_and_stop.pl*) and to perform analyses using the longest isoform per gene (*agat_sp_keep_longest_isoform.pl*).

## Comparative genomics

**Genomes compared.** We compared our assembly and annotation against two previously published *T. circumcincta* assemblies. The first draft genome assembly for *T. circumcincta* was

published in 2017 (Assembly name: T_circumcincta.14.0.ec.cg.pg; assembly accession: GCA_002352805.1) [27], derived from a partially inbred anthelmintic susceptible parasite isolate from New Zealand, hereafter referred to as the WASHU genome. The second assembly was derived from the T_circumcincta.14.0.ec.cg.pg assembly but curated and scaffolded using Hi-C by the DNAzoo Consortium to achieve chromosome-scale scaffolds [27,81,82] (https://www.dnazoo.org/assemblies/Teladorsagia_circumcincta; assembly name: T_circumcincta.14.0.ec.cg.pg_purgehaplotigs_HiC.assembly). These three *T. circumcincta* assemblies were also assessed against the chromosomal assembly of the *H. contortus* ISE isolate [69] and the *C. elegans* N2 reference genomes.

**Repeat Content.** Repetitive sequences were assessed in the three *T. circumcincta* genomes. For each genome, *RepeatModeler* v2.0.1 was used to detect and classify repeats using default parameters, followed by *RepeatMasker* 4.1.0 (parameters: -s -html -gff -small -poly) to annotate the genome and summarise the repeat classes. Repeat density per Mb was calculated using a custom script and plotted by chromosome (**Fig G A in S1 Appendix**).

**Genome completeness.** A comparison of intermediate and complete assembly versions was performed using *BUSCO* v.5.6.1 [83] (parameter:—genome) using the nematoda_odb10 lineage reference datasets. Completed assemblies with annotations were further assessed by *BUSCO* using predicted translated proteins (parameter:—protein). Genome assemblies were also evaluated using *compleasm* [84]. Gene density per Mb was calculated using a custom script and plotted by chromosome (**Fig G B in S1 Appendix**). In addition, *compleasm* output was used to plot the number of busco genes per Mb, indicating whether they were identified as a single copy or duplicated in the tci2_wsi3.0 assembly (**Fig G C in S1 Appendix**).

**Protein comparisons.** Protein sequence orthology between genome assemblies was assessed using *Orthofinder* v.2.3.3 [85]. Predicted protein sequences were obtained from the genome and annotation using *gffread*, from which the longest protein sequence per gene was chosen to assess using *Orthofinder*. To enhance the comparison between WASHU and tci2_wsi3.0 assemblies, we provide the orthogroup mapping between predicted protein sequences from the two assemblies in **S11 Table**. Functional annotations for the protein sequences were inferred using *InterProScan* v.5.57–90.0 against all databases (—goterms—iprlookup). The output of *InterProScan* was collated, and the functional annotations were added to the GFF using *AGAT* [80] *agat_sp_manage_functional_annotation.pl*. To support further analysis using these data, we provide GO terms associated with tci2_wsi3.0 transcripts in **S12 Table**.

## Population genomic analyses

**Parasite material for population genetic analyses.** Two UK field populations of *T. circumcincta* were obtained from commercial sheep flocks to investigate genetic markers of resistance using the faecal egg count reduction test (FECRT) in 2015 and 2016. Here, resistance is defined as a reduction in arithmetic mean faecal egg count (FEC) of less than 95%, with the lower 95% CI less than 90% [86]. Farm 1, sampled in August 2016, was located in SW England and has been previously described [87,88]. This flock ('Farm 8' in Glover et al. [88]) had resistance to multiple anthelmintic classes diagnosed in 2013; $FECRT_{benzimidazole}$ = 83.7%, $FECRT_{levamisole}$ = 92.5%, and $FECRT_{ivermectin}$ = 93.0% reduction in faecal egg output. Subsequently, a second flock was introduced to Farm 1 to form a single, larger flock. Following this, for the current study, a FECRT was performed in 2016. Briefly, sheep were individually weighed to the closest 0.5 kg before being treated with 0.2 mg/kg ivermectin (Noramectin 0.08% oral solution for sheep, Norbrook, UK), with the dose rounded up to the nearest 0.2 ml [87]. Eggs were counted using a modified Mini-Flotac method, with a detection limit of five eggs per gram

(epg) [87]. For Farm 2, located in central Scotland, a FECRT was performed in July 2015. Sheep were treated with 5 mg fenbendazole/kg (10% Panacur, MSD Animal Health Ltd, UK), 0.2 mg ivermectin/kg (Oramec, Boehringer Ingelheim Animal Health, UK), and 7.5 mg levamisole hydrochloride/kg (3% Levacur, MSD Animal Health Ltd, UK), being individually weighed and dosed according to weight, and a FEC was performed using a cuvette method with a detection limit of 1 epg [89]. On both farms, faeces were collected pre- (on the day of treatment) and 14 days post-ivermectin treatment of sheep, and L3 were recovered following coproculture; for Farm 1, faeces were stored at 8˚C for one week and then cultured at 20˚C for 14 days, while for Farm 2 faeces were cultured at 20˚C for ten days. For this study, individual L3 were identified at the species level using PCR of the ITS2 region, and the mean FECs were adjusted to calculate an approximate percentage faecal egg count reduction (FECR%) for *T. circumcincta* as previously described [90]. Arithmetic means and standard deviations were calculated using Microsoft Excel. The *eggCounts* R package v2.4 [91], *fecrt_stan*, with the unpaired model was used to calculate the FECR% and highest posterior density interval. This model was chosen as it allowed unequal group sizes and individual correction factors for each FEC. For Farm 1, the model accounted for zero inflation, arising from low counts that fall below the FEC detection limit and are falsely observed as zero. For Farm 2, the FEC method had a detection limit of 1 epg, hence, no egg counts were taken to be genuinely zero, and zero inflation adjustment was not used. Data for the FECRT are shown in **S1 Table** and **Fig C in S1 Appendix**.

A collection of archival samples biobanked at the Moredun Research Institute were also sequenced. These samples included two UK *T. circumcincta* isolates previously characterised for their susceptibility to drug treatment; MTci1, isolated in 1979 from Moredun Research Institute pastures (drug-sensitive, although suspected low-levels of benzimidazole resistance) and MTci2, the same isolate as used for the genome assembly. Two multi-drug-resistant isolates were also sequenced. MTci5 (benzimidazole, levamisole and ivermectin-resistant) was originally isolated in 2002 from a farm in central Scotland following a diagnosis of triple anthelmintic resistance in an experimental Suffolk flock a year previously [92,93], with subsequent diagnosis of a lack of persistency of moxidectin in a commercial ewe flock grazed on the same pastures [94]. MTci7 (benzimidazole, levamisole, ivermectin, and moxidectin-resistant) was isolated from a large upland sheep farm in Scotland in 2007 following diagnosis of moxidectin resistance by FECRT, subsequent to detection of a lack of persistency of moxidectin two years previously [95]. All samples consisted of hundreds of adult or L4 worms, sampled before or after drug treatment, as described in **S2 Table.**

**Illumina library preparation and sequencing.**   Genomic DNA was extracted from the UK field populations (Farm 1 and Farm 2). Initially, DNA lysates of individual larvae were prepared using DirectPCR Lysis Reagent (Cell; Viagen Biotech) supplemented with DTT (Thermofisher Scientific) and Proteinase K (Fungal; Invitrogen), and speciation PCRs performed using species-specific ITS2 single- and multi-plex PCRs to identify *T. circumcincta* L3 from other strongyle larvae, as previously described [96]. Ninety-one *T. circumcincta* L3 were pooled, using half of the total lysate volume, and genomic DNA was extracted using a phenol-chloroform method supplemented with glycogen during the final ethanol precipitation to enhance the recovery of the low DNA quantity present, as previously described [90]. The final precipitated genomic DNA was resuspended in 10 µl of EB buffer (Invitrogen). Illumina sequencing libraries were prepared, followed by 75 bp paired-end sequencing of each library on an individual lane of Illumina HiSeq 4000. For Farm 2, a technical replicate of genomic DNA extraction, library preparation and sequencing was performed, and these data are referred to as 2A and 2B (**S2 Table** and **Fig E in S1 Appendix**).

To assess linkage disequilibrium, we used double digest restriction-site associated DNA sequencing (ddRAD-seq) on sixty-two individual L3 (30 pre- and 32 post-ivermectin treated L3) from Farm 1. These L3 were a subset of those included within the larger number pooled for whole genome sequencing above. Briefly, half the volume of DNA from individually lysed larvae (DirectPCR Lysis Reagent (Cell; Viagen Biotech), DTT, Proteinase K) was whole genome amplified (REPLI-g Single Cell Kit, Qiagen) before digestion with two restriction endonucleases, MluCI and NlaIII (New England Biolabs), as previously described [90]. Barcoded adaptors [97] (**S3 Table**) were ligated to 250 ng of digested whole genome amplified DNA. Pooled sample DNA libraries were size selected (225–325 bp) using a Pippen Prep (Sage Science), followed by the addition of Illumina adapters by PCR using KAPA HiFi (KAPA Biosystems) PCR for eight cycles. Genomic libraries were sequenced with 125 bp paired-end reads on an Illumina HiSeq 2000v4.

Genomic DNA was extracted from the archival drug-resistant isolates using the Isolate II DNA extraction kit following the manufacturer's instructions, with a final elution in 50 µl buffer. Illumina sequencing libraries were prepared, followed by 150 bp paired-end sequencing on a single lane of Illumina NovaSeq 6000 (**S2 Table**).

**Mapping Illumina reads to reference genomes.**   In addition to the sequencing data generated for this study and described above, we also reanalysed pooled sequencing data from a semi-inbred drug-susceptible ($S_{inbred}$) and a multidrug-resistant, backcrossed isolate ($RS^3$) originally derived from a resistant field isolate; the founding isolates were collected from farms in New Zealand. These data were originally generated to map drug resistance variation as described by Choi and colleagues [27] and are available from ENA BioProject PRJNA72569 ($RS^3$: SRR4026813; $S_{inbred}$: SRR4027077) (**S2 Table**).

Sequenced and ENA-derived Illumina reads were quality checked using *FastQC* v0.12.1 (https://www.bioinformatics.babraham.ac.uk/projects/fastqc/) and *MultiQC* v1.17 [98]. Sequencing reads were mapped to a reference genome using a *Nextflow* mapping pipeline (*mapping-helminth/v1.0.8*). Briefly, the pipeline first converted raw reads into an unmapped BAM (uBAM) file using *FastqToSam* and subsequently processed them using *MarkIlluminaAdapters* from *GATK* v.4.1.4.1 before using *minimap2* v.2.16 [99] to map the sequencing reads to the *T. circumcincta* tci2_wsi3.0 reference genome. Mapped reads were sorted, and duplicate reads were marked using *samtools* and *sambamba* [100], respectively. Mapping statistics were generated using *samtools flagstats* and summarised using *MultiQC*.

**Comparison of coverage between farm samples over chromosome 5.**   To compare coverage between pre- and post-ivermectin samples along chromosome 5, the depth per site was calculated using *samtools* v1.17 *depth* (-q 30 -Q 30 -a), requiring both the base and mapping quality to be at least 30, and for all sites to be included in the output, even if zero coverage, from which the average depth was calculated in 100 kb windows along the chromosome. Note that split and secondary hits had already been removed using a combination of *sambamba* and *awk*. Next, the coverage was standardised by using the median chromosome 5 coverage. Finally, the ratio of the post-ivermectin:pre-ivermectin sample was calculated.

**Within and between sample analyses of genetic differentiation.**   To calculate between-sample genetic differentiation, we used *Grenedalf* v0.3.0 *fst* [101] to calculate an unbiased-Nei measure of $F$st in 100 kb sliding windows. We ran calculations separately for the "farm", "choi", and "strain" data using mapped bam files as input, keeping the run parameters consistent (--method unbiased-nei --sam-min-map-qual 30 --sam-min-base-qual 30 --filter-sample-min-count 2 --filter-sample-min-coverage 10 --filter-sample-max-coverage 100 --window-type sliding --window-sliding-width 100000 --write-pi-tables) but to account for the different estimated pool sizes of each sample group (--pool-sizes = 800, 180, and 200 for the "choi", "farm", and "strain" analyses, respectively). Pairwise Fst data per 100 kb for all sample

combinations within each of the "farm", "choi", and "strain" groups are presented in **S13, S14, and S15 Tables**, respectively.

To assess the impact of coding sequence variation and determine individual variant frequencies, variants were identified using *bcftools* (v1.14) *mpileup* (-B -q 30 -Q 30 --ff DUP -d 100) followed by *bcftools call* (-mv -O v). Putative coding variation was assessed using *SnpEff* (v5.2a) [102].

To identify whether any SNP alleles were under selection by ivermectin on both UK farms, a Cochran-Mantel-Haenszel test was performed using *PoPoolation* 2 *cmh-test.pl* (v1201, "popoolation2-code" version which provided the log odds ratio) (--min-count 3 --min-coverage 10 --max-coverage 2%--population 1–2,3–5). The Farm 1 pre- and post-treatment pairwise comparison was compared with the Farm 2A pre- and post-treatment pairwise comparison. P values were filtered using both the FDR (Benjamini-Hochberg), choosing an FDR of 5%, 1% and 0.01%, and also by using the Bonferroni method. Finally, the proportion of SNPs ranked within the top 1% of the lowest p-values, which also had a probability of a reduction in the reference allele in the post-treatment samples relative to the pre-treatment samples (log odds ratio < 1), were calculated in 1 Mb windows along the chromosomes and plotted using *ggplot2*.

**Gene copy number variation.** To determine the relative copy number difference of genes between susceptible and resistant isolates, we first determined the read coverage of all CDS exons within the annotation using *bedtools* (v2.29.0) *multicov*. In *R*, the median CDS coverage per gene was determined, from which a ratio of resistant-to-susceptible gene coverage was calculated. To determine outliers of coverage ratio, thereby identifying genes with significantly higher or lower coverage in the resistant relative to susceptible isolate, we ranked genes by the susceptible isolate coverage and plotted the coverage of the resistant isolate using this gene order. This approach was used to assess copy number variation in *pgp-9* (TCIR_00132080).

**ddRAD-seq analyses.** Sequencing data were de-multiplexed using *Stacks* v2.5 *process_rad-tags* (-r -q), followed by mapping to the tci2_wsi3.0 assembly using the *mapping-helminth* v1.0.8 pipeline described above. Variants were called using *GATK HaplotypeCaller* v4.1.4.1 (parameters: -ERC GVCF --min-base-quality-score 30 --disable-read-filter NotDuplicateReadFilter) per sample, followed by *GATK CombineGVCFs* to compile all 62 sample sets into a single vcf file. The vcf was filtered using *GATK VariantFiltration* to keep only variants successfully genotyped in at least 30 individuals (--filter-expression "AN < 60") and with a mapping quality greater than 30 (--filter-expression "MQ < 30"). A threshold of 30 individuals was chosen to allow for the inclusion of RAD sites, which were present in the pre-treatment or the post-treatment sample population but lost in the other. A prior analysis using an earlier version of the assembly had identified that two individuals–one in the pre-ivermectin sample and the other in the post-ivermectin sample–were identical, and thus, these individuals were removed from further analysis (see [90] for more details).

Linkage disequilibrium within each sample was analysed along chromosome 5 and between chromosomal scaffolds using *PLINK* v1.9 [103]. Inter-chromosomal $r^2$ was calculated using *plink --r2 inter-chr with-freqs*, allowing 25% missing individuals (--geno 0.25) and a minimum maf (--maf 0.05). Sites were filtered only to include one variant every 100 kb (--bp-space 100000), and all $r^2$ values were retained (--ld-window-r2 0). To calculate $r^2$ along chromosome 5 (--chr tci2_wsi3.0_chr_5), the same command was used (--r2 with-freqs), but increasing the number of sites to a maximum of one site every 1 kb (--bp-space 1000), and setting the maximum 'distance' between two SNPs at 1 Mb (--ld-window-kb 1000) or 5 Mb, and with no more than 999 other SNPs between them (--ld-window 1000). Finally, the output data were restricted to only $r^2$ values of 0.001 or higher. The decay of linkage disequilibrium over chromosome 5 was investigated. For this, a script written by Fabio Marroni was used [104]. The

distance between variants (in bp) to the half-decay value ($r^2 = 0.5$) was calculated. Next, the distance to the mean sample inter-chromosomal $r^2$ value was calculated. As these were slightly different for the pre- and post-ivermectin samples, the distance to an $r^2$ value of 0.05 was also calculated.

## Supporting information

**S1 Appendix. The file contains supplementary text and figures, including the following: Note on monepantel and genes associated with resistance.** Fig A. Comparison of chromosomal scaffolds between tci2_wsi3.0 and DNAZOO genome assemblies. Fig B. Genetic differentiation between all samples. Fig C. Farm mean faecal egg count reduction test results, adjusted by L3 species proportions. Fig D. Distinguishing genetic drift from selection on the UK Farm populations. Fig E. Comparison of genetic variation between technical replicates. Fig F. Relative sequencing coverage ratio along chromosome 5 between Farm 1 post:pre, Farm 2A post:pre and between Farm 1 post:Farm 2A post. Fig G. Gene, BUSCO, and repeat density. Fig H. Proportion of SNPs in 1 Mb windows identified as 'significant' by Cochran-Mantel-Haenszel test. Fig I. Genome-wide gene coverage between susceptible and resistant isolates. (DOCX)

**S1 Table. Faecal egg count reduction test results from UK field populations.** (XLSX)

**S2 Table. Sequencing overview.** (XLSX)

**S3 Table. ddRAD-seq barcodes using for demultiplexing, reads, and mapping rates.** (XLSX)

**S4 Table. Genome and proteome completeness assessed using BUSCO and *compleasm*.** (XLSX)

**S5 Table. Repeat content of the *Teladorsagia circumcincta* genomes.** (XLSX)

**S6 Table. Number of shared one-to-one orthologs between proteomes.** (XLSX)

**S7 Table. Candidate genes and variants associated with anthelmintic resistance.** (XLSX)

**S8 Table. Haplotype frequency of susceptible and resistant beta-tubulin isotype 1 variants.** (XLSX)

**S9 Table. Genes in the ivermectin-associated chromosome 5 peak.** (XLSX)

**S10 Table. WormBase ParaSite v18 BIOMART information for orthologues of genes in the ivermectin-associated chromosome 5 peak.** (XLSX)

**S11 Table. Orthogroup mapping between tci2_wsi3.0 and WASHU predicted proteins.** (XLSX)

**S12 Table. GO terms for tci2_wsi3.0 predicted protein IDs.** (XLSX)

**S13 Table. Fst data per 100 kb window for all pairwise comparison of Farm samples.** (XLSX)

**S14 Table. Fst data per 100 kb window for pairwise comparison of Choi samples.** (XLSX)

**S15 Table. Fst data per 100 kb window for all pairwise comparison of Strain samples.** (XLSX)

## Acknowledgments

We would like to acknowledge members of the BUG Consortium, and the Parasite Genomics and Helminth Genomics groups at the Wellcome Sanger Institute for insightful discussions throughout this project. We also thank Pathogen Informatics and DNA Pipelines (WSI) for their support and expertise and the Biosciences Division at the Moredun Research Institute for expert care and animal assistance. We gratefully acknowledge assistance from participating farmers and their veterinarians. For the purpose of Open Access, the authors have applied a CC BY public copyright Licence to any Author Accepted Manuscript version arising from this submission.

## Author Contributions

**Conceptualization:** James A. Cotton, Eileen Devaney, Roz Laing, Stephen R. Doyle.

**Data curation:** Jennifer McIntyre, Stephen R. Doyle.

**Formal analysis:** Jennifer McIntyre, Stephen R. Doyle.

**Funding acquisition:** Alasdair J. Nisbet, Tom N. McNeilly, Dave Bartley, Matt Berriman, James A. Cotton, Eileen Devaney, Roz Laing, Stephen R. Doyle.

**Investigation:** Jennifer McIntyre, Alison Morrison, Kirsty Maitland, Duncan Berger, Daniel R. G. Price, Alan Tracey, Katie Bull, Hannah Rose Vineer, Eric R. Morgan, Dave Bartley, Stephen R. Doyle.

**Methodology:** Stephen R. Doyle.

**Project administration:** Nancy Holroyd, Matt Berriman, James A. Cotton, Eileen Devaney, Roz Laing, Stephen R. Doyle.

**Resources:** Alison Morrison, Kirsty Maitland, Duncan Berger, Daniel R. G. Price, Dionysis Grigoriadis, Katie Bull, Hannah Rose Vineer, Eric R. Morgan, Alasdair J. Nisbet, Tom N. McNeilly, Yvonne Bartley, Neil Sargison, Dave Bartley, Matt Berriman, Eileen Devaney.

**Software:** Jennifer McIntyre, Sam Dougan, Dionysis Grigoriadis, Alan Tracey, Stephen R. Doyle.

**Supervision:** Dave Bartley, James A. Cotton, Eileen Devaney, Roz Laing, Stephen R. Doyle.

**Visualization:** Jennifer McIntyre, Stephen R. Doyle.

**Writing – original draft:** Jennifer McIntyre, Stephen R. Doyle.

**Writing – review & editing:** Jennifer McIntyre, Alison Morrison, Kirsty Maitland, Duncan Berger, Daniel R. G. Price, Sam Dougan, Dionysis Grigoriadis, Alan Tracey, Nancy Holroyd, Katie Bull, Hannah Rose Vineer, Mike J. Glover, Eric R. Morgan, Alasdair J. Nisbet, Tom N. McNeilly, Yvonne Bartley, Neil Sargison, Dave Bartley, Matt Berriman, James A. Cotton, Eileen Devaney, Roz Laing, Stephen R. Doyle.

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
