## [Decision Letter · Decision Letter 0]

3 Sep 2024

Dear Dr Doyle,

Thank you very much for submitting your manuscript "Chromosomal genome assembly resolves drug resistance loci in the parasitic nematode Teladorsagia circumcincta" for consideration at PLOS Pathogens. As with all papers reviewed by the journal, your manuscript was reviewed by members of the editorial board and by several independent reviewers. In light of the reviews (below this email), we would like to invite the resubmission of a significantly-revised version that takes into account the reviewers' comments.

All reviewers hold a positive view of the contributions of this manuscript and offer mostly minor but useful suggestions to improve technical clarity. I classified this as a 'major revision' only to account for the larger number of suggestions and constructive critiques provided by reviewer 2. I look forward to receiving a revised manuscript addressing these comments.

Sincerely,

Mostafa Zamanian

Academic Editor

PLOS Pathogens

James Collins III

Section Editor

PLOS Pathogens

Michael Malim

Editor-in-Chief

PLOS Pathogens

orcid.org/0000-0002-7699-2064

All reviewers hold a positive view of the contributions of this manuscript and offer mostly minor but useful suggestions to improve technical clarity. I classified this as a 'major revision' only to account for the larger number of suggestions and constructive critiques provided by reviewer 2. I look forward to receiving a revised manuscript addressing these comments.

Reviewer's Responses to Questions

**Part I - Summary**

Reviewer #1: This is a clearly written, well presented, and scientifically rigorous description of the genomic analysis of anthelmintic resistance in Teladorsagia circumcincta. It should be accepted for publication in PLoS Pathogens. I have made some comments below, separated into those to which I think the authors should respond prior to final acceptance, and those that are of lesser importance.

As is clear from my comments, but I will state here as well, I am a co-author of the Choi et al. paper to which this paper refers extensively and from which the authors have extracted and re-analysed data. Some of my comments relate specifically to the original paper and the study it reported.

Reviewer #2: (No Response)

Reviewer #3: This manuscript describes a very much-improved Teladorsagia circumcincta reference genome constructed using PacBio long reads and Hi-C contact maps. This new resource enabled the mapping and further refinement of anthelmintic resistance loci, vastly improving results for this species, as the authors had done previously for H. contortus. This manuscript describes a much-needed community resource.

**Part II – Major Issues: Key Experiments Required for Acceptance**

Reviewer #1: • Lines 345-68 & Fig 2: BZ resistance. An intriguing feature of the Choi et al. analyses was the absence of tubulin isotype 1 E198L / F200Y and double homozygotes and the consequent strong departure from Hardy-Weinberg equilibrium at this locus. The authors speculated that the this implied that the resistance alleles in heterozygotes were always in trans- rather than cis- on the basis that individuals homozygous for both resistance alleles were unviable. The highly fragmented nature of the Choi et al. assembly did not, however, permit testing of this hypothesis through haplotype reconstruction. This manuscript include very extensive PacBio sequencing, including sequencing from DNA made up of large pools (assuming that the 125K sheathed L3 referred to in Supp Table 2, for example, refers to 125,000 individuals, or 250,000 haploid genomes, per sequencing pool). Each PacBio read is, in effect, a haplotype. The inter-strain comparisons and GWAS are all based on measures that use allele frequency from genotyping of bi-allelic SNP loci. Could these authors please consider the value of considering haplotypes given that they have data that may permit haplotype imputation from the haplotype frequencies in the strain MTci2 PacBio data? It is worth noting that most of the genetic variation between individuals is haplotype variation generated by recombination, and it is on haplotypes rather than individual alleles at single loci, that selection will operate.

• Lines 400 – 410: in reference to the comment above concerning the importance of haplotypes, I note that lines 400 – 410 touch on the importance of genetic linkage of loci that may be under selection i.e., an issue of haplotypes rather than single alleles at single loci. As with old-fashioned Mendelian genetic mapping by recombination, perhaps looking for the rare recombinant that separates linked alleles at two loci involved in different resistances (in this case, BZ vs LEV) may be the way to go?

• Lines 491 – 510: Tcpgp-9 variation. This paragraph describes data consistent with a causal involvement of increased copy number of the Tcpgp-9 locus in resistance to ivermectin in some of the lines analysed. The authors propose that this is unlikely to be a major driver of ivermectin resistance on the grounds of “The relative height of the pgp-9 peak to other peaks…”. I take this to be a reference to an Fst peak height (there is no specific reference to a figure here; line 507). Could these authors please explain in more detail why there should be a direct linear correlation between the copy number of a specific locus and Fst? Particularly if, as seems plausible for copy number variation in response to recently applied drug selection, the copy number expansion is recent and there is little genetic variation amongst the duplicated copies. One could imagine a small degree of sequence variation (measured by Fst) despite recent 4 – 10 fold expansion of copy number (Fig 4G).

• Lines around and preceding 578 – 579. The authors make this more complicated than it needs to be and are possibly inadvertently misleading the reader by referring to “these shared signals of selection” when trying to express the view that (a) the mechanisms of BZ and IVM resistance are distinct, (b) selection for BZ resistance preceded selection for IVM resistance so that (c) worms are frequently resistant to both classes of drug, and thus display two mechanistically and genetically distinct signals of selection. The challenge is to determine which signa(s) relate to which resistance. The authors would do well to point to the Choi et al. strategy of using genetic recombination and single drug challenge to separate the signals for each drug class. There was little “shared signal” between drug classes in the F3 recombinant generation of the Choi study. I agree that attempting to reconstruct selection history through haplotype analysis (lines 575-85) may be necessary given the impracticality of repeating the Choi et al. genetic experiment (which took a number of years to complete), but the authors might consider attempting to rewrite this section of the Discussion. As written, it does not clearly annunciate the two related, but separate, routes to solving the problem that it is difficult to assign a peak of genetic differentiation to a specific drug resistance in multi-resistant strains: the genetic solution presented by Choi et al. vs the theoretical solution of reconstructing the history of haplotype selection.

• Line 609 onwards: the authors might consider adding text to address the following to this paragraph. First, in reference to lines 609 – 11, it is clear that to be useful in a routine diagnostic sense, identifying these conserved pathways to resistance is essential. Do the authors see any potential here to discuss how identifying these conserved pathways might lead to practical outcomes? If not, why not? Second, and related to the first, the authors conclude that there “resistance [to BZ and IVM] has evolved independently on diverse haplotypes”. I agree, but how likely is this to impede the development of diagnostic tools based on the shared resistance mechanism (the endpoint of selection) even if the genetic history of selection is unique in each resistant population? What are the practical consequences? The authors may respond by saying that the aim of their work is provide better tools to investigate the genetic basis of resistance but not to develop tools for genetic diagnosis of resistance. I remind them that the final sentence states clearly that better genome data will “inform more effective and sustainable control” by identifying “the specific causal variants driving resistance”. Those variants are known for tubulin isotype-1 but no diagnostic tools are in routine use. How will providing analogous information on IVM and/or LEV resistance help? Very non-molecular work such as that described by Leathwick and others on how to use anthelmintics to delay and perhaps even revert resistance has provided practical answers.

Reviewer #2: (No Response)

Reviewer #3: I have no major issues.

**Part III – Minor Issues: Editorial and Data Presentation Modifications**

Reviewer #1: • Line 40-41 (abstract): reads “most important pathogens of [sheep and goat] farming”, which implies that T. circ is a pathogen of farming. Suggest delete “farming” from this sentence to read simply “most important pathogens of sheep and goats”. This is repeated in lines 119-120 in the main text.

• Line 572: Would “concurrent” be more correct? According to the online Merriam-Webster Dictionary, intercurrent is a term used in pathology in reference to a second, unrelated, disease that occurs during the course of a first or initial condition. The authors mean, I think, combination anthelmintic therapies. Why not say that BZ’s are used frequently in combination formulations with ML’s? That is what happens. Or do the authors mean that IVM (and/or pther ML’s) are used primarily when BZ’s fail? Or simply that since BZ’s preceded IVM into the market by two decades, extensive selection for BZ resistance had occurred before IVM was introduced?

Reviewer #2: (No Response)

Reviewer #3: I have only three minor issues.

1 --- The comparisons in Figure 2 are difficult to interpret. I had to go back and forth between the text and the figure, which made the interpretations more difficult. I suggest that the colors for different chromosomes can be dropped. They do not add any meaning that is not already conveyed by the plotting in separate boxes for linkage groups. I suggest that colors could be used to show which drugs were used in the different isolates in the comparisons. If that ends up being too confusing, then maybe expanding the facet labels would be helpful.

2 --- How many beta-tubulin genes are found in this new assembly? As seen in other nematode species, the composition of beta-tubulin can influence benzimidazole resistance. These data could just be an interesting point to add along with the beautiful connections to putative isotype 1 and isotype 2 variation in Teladorsagia circumcincta BZ resistant isolates.

3 --- The authors state that moxidectin and ivermectin are similar, which they are as MLs. However, the Kaplan lab has shown that IVM resistance and MOX resistance are likely different because ordered treatments lead to different results in H. contortus. Milbemycins like MOX can circumvent IVM resistance for a period of time before general ML resistance is detected by FECR and LDA. The section on MOX just needs some slight edits to clarify this point.

PLOS authors have the option to publish the peer review history of their article (what does this mean?). If published, this will include your full peer review and any attached files.

Reviewer #1: **Yes: **Warwick Grant

Reviewer #2: No

Reviewer #3: No
---

## [Decision Letter · Decision Letter 1]

9 Dec 2024

Dear Dr Doyle,

We are pleased to inform you that your manuscript 'Chromosomal genome assembly resolves drug resistance loci in the parasitic nematode Teladorsagia circumcincta' has been provisionally accepted for publication in PLOS Pathogens.

Best regards,

Mostafa Zamanian

Academic Editor

PLOS Pathogens

James Collins III

Section Editor

PLOS Pathogens

Sumita Bhaduri-McIntosh

Editor-in-Chief

PLOS Pathogens

orcid.org/0000-0003-2946-9497

Michael Malim

Editor-in-Chief

PLOS Pathogens

orcid.org/0000-0002-7699-2064

Reviewer Comments (if any, and for reference):

Reviewer's Responses to Questions

**Part I - Summary**

Reviewer #2: All concerns raised during the review process were addressed satisfactorily in the revised version of the manuscript. The authors provided clear and thorough responses to all comments, incorporating the suggested changes where applicable and offering detailed justifications for their approach. The revisions significantly improved the clarity, rigor, and overall quality of the manuscript, ensuring that the final version aligns with the standards of the field.

Reviewer #3: The authors have sufficiently addressed my comments.

**Part II – Major Issues: Key Experiments Required for Acceptance**

Reviewer #2: (No Response)

Reviewer #3: (No Response)

**Part III – Minor Issues: Editorial and Data Presentation Modifications**

Reviewer #2: (No Response)

Reviewer #3: (No Response)

PLOS authors have the option to publish the peer review history of their article (what does this mean?). If published, this will include your full peer review and any attached files.

Reviewer #2: No

Reviewer #3: No

---

## [Editor Report · Acceptance letter]

4 Jan 2025

Dear Dr Doyle,

We are delighted to inform you that your manuscript, "Chromosomal genome assembly resolves drug resistance loci in the parasitic nematode Teladorsagia circumcincta," has been formally accepted for publication in PLOS Pathogens.

Best regards,

Sumita Bhaduri-McIntosh

Editor-in-Chief

PLOS Pathogens

orcid.org/0000-0003-2946-9497

Michael Malim

Editor-in-Chief

PLOS Pathogens

orcid.org/0000-0002-7699-2064